# Global urban fractional changes at a 1km resolution throughout 2100 under eight SSP-RCP scenarios

Wanru He[1], Xuecao Li[1,2]*, Yuyu Zhou[3,4]*, Zitong Shi[5], Guojiang Yu[1], Tengyun Hu[6], Yixuan Wang[1], Jianxi Huang[1,2], Tiecheng Bai[7], Zhongchang Sun[8], Xiaoping Liu[9], Peng Gong[3,4]

[1] College of Land Science and Technology, China Agricultural University, Haidian, Beijing 100083, China
[2] Key Laboratory of Remote Sensing for Agri-Hazards, Ministry of Agriculture and Rural Affairs, Beijing 100083, China
[3] Department of Geography, The University of Hong Kong, Hong Kong 999077, China
[4] Urban Systems Institute, The University of Hong Kong, Hong Kong 999077, China
[5] National Institute of Natural Hazards, Ministry of Emergency Management of China, Beijing 100085, China
[6] Beijing Municipal Institute of City Planning and Design, Beijing 100045, China
[7] School of Information Engineering, Tarim University, Alaer 843300, China
[8] Key Laboratory of Digital Earth Science, Aerospace Information Research Institute (AIR), Chinese Academy of Sciences,
Beijing 100101, China
[9] School of Geography and Planning, Sun Yat-Sen University, Guangzhou 510275, China
*Correspondence to*: Xuecao Li (xuecaoli@cau.edu.cn) and Yuyu Zhou (yuyuzhou@hku.hk)

**Abstract.** The information of global spatially explicit urban extents under scenarios is important to mitigate future

environmental risks caused by global urbanization and climate change. Although future dynamics of urban extent were

commonly modelled with conversion from non-urban to urban using cellular automata (CA) based models, gradual changes

of impervious surface area (ISA) at the pixel level were limitedly explored in previous studies. In this paper, we developed a

global dataset of urban fractional changes at a 1km resolution from 2020 to 2100 (5-year interval), under eight scenarios of

socioeconomic pathways and climate change. First, to quantify the gradual change of ISA within the pixel, we characterized

ISA growth patterns over past decades (i.e., 1985-2015) using a sigmoid growth model and annual global artificial impervious

area (GAIA) data. Then, by incorporating the ISA-based growth mechanism with the CA model, we calibrated the state-

specific urban CA model with quantitative evaluation at the global scale. Finally, we projected future urban fractional changes

at 1km resolution under eight development pathways based on the harmonized urban growth demand from Land Use

Harmonization2 (LUH2). The evaluation results show that the ISA-based urban CA model performs well globally, with an

overall $R^2$ of 0.9 and the Root Mean Square Error (RMSE) of 0.08 between modeled and observed ISAs in 2015. With the

inclusion of temporal contexts of urban sprawl gained from GAIA, the dataset of global urban fractional change shows good

agreement with 30-year historical observations from satellites. The dataset can capture spatially explicit variations of ISA and gradual ISA change within pixels. The dataset of global urban fractional change is of great use in supporting quantitative analysis of urbanization-induced ecological and environmental change at a fine scale, such as urban heat islands, energy consumption, and human-nature interactions in the urban system. The developed dataset of global urban fractional change is

available at https://doi.org/10.6084/m9.figshare.20391117.v4 (He et al., 2022).

**1 Introduction**

Spatially explicit global urban extents and dynamics play essential roles in assessing future urbanization-induced environmental and climate impacts (Acuto et al., 2018; Castán Broto and Bulkeley, 2013). The latest World Urbanization Prospects (Nations et al., 2019) reveal that the global urban population has grown rapidly from 750 million in 1950 to 4.5

billion in 2021, and it is anticipated to reach around 7 billion by 2050, with more than 2 billion people migrating from rural to urban areas. Consequently, the urban expansion growth rates have been higher over past decades in rapidly developing regions (e.g., China and India), leading to a notable urban sprawl with rising concerns in the urban environment (Güneralp et al., 2017; Alberti et al., 2017), energy consumption (Li et al., 2019c), and human-environmental interactions. As such, the realization of sustainable development goals (SDGs) is challengeable due to potential urbanization-induced threats from various

environmental issues, such as agriculture land loss (Seto and Ramankutty, 2016), deforestation (Foley et al., 2005; Defries et al., 2010), and air pollution (Gong et al., 2012). Long-term urban extent dynamics are attributable to revealing different urbanization stages, which can notably reduce uncertainties of future urban sprawl pathways and are of great potential to support further decision-making on global urban and environmental changes (Klein Goldewijk et al., 2010).

Temporal contexts of urban sprawl have not been comprehensively used in cellular automata (CA)-based models, although

various urban CA models have been developed to model urban sprawl in a spatially explicit manner (Li and Gong, 2016). Presently, there are different CA-based models have been developed for various applications, such as the Logistic-CA model (B. Wu, Huang, & Fung, 2009), the Fuzzy-CA model (Liu, 2012; Liu and Phinn, 2003), the Agent-based CA model (Li and Liu, 2007; Liu et al., 2008), and the Patch-based CA model (Chen et al., 2014). Given that the neighborhood is a crucial

component in the urban CA model, many relevant studies have been conducted for improvement by configuring the shape and size of the neighborhood in a spatially explicit manner (Santé et al., 2010; Chen et al., 2014; Li et al., 2014; Kocabas and Dragicevic, 2006; Wu et al., 2012). For instance, the impact of neighborhood configuration (e.g., distances and land use/cover compositions) has been widely explored in many studies for model development (Wu et al., 2012; Wu et al., 2019; Liao et al., 2016; Dahal and Chow, 2015). Unfortunately, most of them mainly focused on the spatial configuration of the neighborhood, whereas the temporal contexts of urban sprawl revealed from long-term and continuous urban extent time series data were hardly been investigated, especially at the global scale, although several studies have confirmed that those newly developed urban pixels have a greater impact on the model performance than those developed in early years (Li et al., 2020; Liu et al., 2017).

Although several global datasets of urban extent dynamic with conversions from non-urban to urban have been proposed, there is still limited effort to characterize the gradual urban fractional change (i.e., ISA) within each grid when projecting future global urban sprawl (Potere et al., 2009; Huang et al., 2021; Herold et al., 2003; Seto et al., 2012; Li et al., 2017). However, the spatial resolution of these global urban products is either relatively coarse (8km) (Gao and O'neill, 2020) or only available in binary format (1km) (Zhou et al., 2019; Chen et al., 2020a). The temporal contexts of urban sprawl have been limitedly considered in these studies, leading to noticeable uncertainties regarding the projected global urban extent dynamics in the future with long-term historical urban sprawl. Probably due to the absence of long-term and fine resolution annual global urban extent time series data (Li et al., 2015; Shi et al., 2017; Song et al., 2016; Brown De Colstoun et al., 2017), characterizing the temporal pattern of urban sprawl dynamics has not been comprehensively explored, in particularly coupling with urban CA models. Although urban fractional data with detailed impervious surfaces have been developed recently, such as the Global Man-made Impervious Surface (GMIS) data (Brown De Colstoun et al., 2017),, information of long-term urban fractional dynamics is still highly required for urban CA model improvement. For instance, the dynamics of historical urban extents derived from satellite observations can be employed to reveal the allometric growth of urban lands under diverse urbanization levels, which is helpful in estimating future urban demand (Li et al., 2019a). Also, the temporal contexts of urban sprawl can

be used as a proxy in the neighborhood configuration, thereby improving the model performance in spaces (Li et al., 2020). It is worth noting that most developed urban CA models for global application commonly adopted the abrupt conversion (i.e., from non-urban to urban) to represent the urban sprawl process during the modeling, which ignores the difference of urban growth rates across grids (e.g., ≥1km) with diverse urbanization stages (Liu et al., 2018; Santé et al., 2010; Li et al., 2014;

Chen et al., 2002; Verburg et al., 2006). Despite the abrupt conversion in urban CA models has limited impacts when implementing them at a fine resolution (e.g., 30m) and at the local scale, it is inevitable to weaken the model performance in fringe areas of cities at the regional and global scale, in which the spatial resolution of grids is relatively coarse (e.g., ≥1km) (Li et al., 2020; Gao and O'neill, 2019).

To fill this research gap, we developed a global dataset of urban fractional changes (1km) through 2100 (with a 5-year interval)

under eight scenarios, using a ISA-based urban CA models. We first characterized state-specific ISA growth patterns over past decades using historical urban extent data and a sigmoid growth model to capture the gradual change of ISA within each grid. After that, we incorporated the ISA-based growth mechanism with the urban CA model and calibrated the ISA-based urban CA model quantitatively with evaluations at the global scale. Finally, we projected gradual urban fractional changes within 1km grid under eight scenarios using the developed ISA-based urban CA model. The remainder of this paper describes the

adopted data (Section 2), the proposed simulation approach (Section 3), the results and discussion (Section 4), the data availability (Section 5), and concluding remarks (Section 6).

**2 Data**

We used the global artificial impervious area (GAIA) as our primary dataset to capture temporal contexts of urban development (Gong et al., 2019; Gong et al., 2020a). Given that there are currently no long-term urban fractional (i.e., ISA) dynamic products

in high spatial resolution (e.g., 1km) directly obtained from satellite observations (Brown De Colstoun et al., 2017), here we adopted the commonly used strategy through spatial aggregation from high-resolution (e.g., 30m) urban extent data to derive the ISA time series data for modeling. The GAIA data record annual global urban extent (i.e., non-urban and urban) at a 30m resolution, spanning from 1985 to 2018, with overall mean accuracies above 90%. Besides, the derived historical urban extents

from GAIA are temporally consistent (i.e., non-urban to urban) over past decades. To characterize the urban fractional changes at the pixel scale (i.e., 1km), we aggregated 30m binary (i.e., urban and non-urban) results from GAIA into the 1km urban fractional maps (i.e., ISA).

We used the Land Use Harmonization 2 (LUH2) data to derive future urban growth pathways under eight scenarios determined by Shared Socioeconomic Pathways (SSPs) and Representative Concentration Pathways (RCPs). In the LUH2 dataset, the gridded-based scenarios (2015-2100) of multiple land-use types (e.g., urban) were developed by Integrated Assessment Models (IAMs), by jointly considering the socioeconomic development and climate change in the future (Mu et al., 2022; Hurtt et al., 2020). The LUH2 dataset provides detailed fractional information of different land use types within the relatively coarse grid (0.25°×0.25°) globally through 2100 with eight SSP-RCP scenarios. Consequently, the LUH dataset has been widely used in various studies, such as land use cover change and carbon emission (Hong et al., 2021; Friedlingstein et al., 2020; Borrelli et al., 2020; Chen et al., 2020a; Li et al., 2016).

In addition, we collected a variety of spatial proxies to estimate the suitability of potential grids for urban development when modeling future global urban fractional change (Li et al., 2020). These spatial proxies reflect different spatial aspects related to urban sprawl, such as locations (e.g., minimum distance to major worldwide cities), traffic networks (e.g., minimum distance to major roads, minimum distance to highways, and minimum distance to local roads), terrain (e.g., digital elevation model and slope), were used to train the logistic regression model and land constraints (e.g., protected areas) (Table 1). For example, some spatial proxies (e.g., land cover and protected area) were defined as specific constrains (e.g., suitable or not), while terrain and location proxies were directly calculated from the DEM and distance to the nearest roads (or city centers), respectively. These spatial proxies (Fig. S1) were used to reflect the synthesized effect on the suitability (also called transition rule) of urban sprawl within each pixel, according to its biophysical and socioeconomic conditions (Li et al., 2014). Besides, these spatial proxies were normalized at a spatial resolution of 1km before the modeling (Li et al., 2014; Li et al., 2020).

**3 Method**

We developed the dataset of global urban fractional change at a 1km resolution under eight SSP-RCP scenarios, using the developed Logistic-Trend-ISA-CA model (He et al., 2023) (Fig.1). First, we characterized the ISA growth patterns within each pixel using long-term urban extent data. The state-specific ISA growth patterns over the past decades were quantified using the sigmoid growth model, given that cities show allometric growth rates during diverse urbanization stages (Fig.1a). Then, we incorporated the ISA-based growth mechanism with the urban CA model and calibrated the developed urban CA model with evaluations at the global scale quantitatively (Fig.1b). Finally, we projected future urban fractional changes at 1km resolution under eight development pathways at the global scale (Fig.1c). The country-specific future urban area growth (or called urban demands) trends between LUH2 and GAIA were harmonized before the modeling. Details of each component in the proposed framework can be referred to in the following subsections.

## 3.1 Characterization of urban fractional change with urban CA model

We characterized urban fractional change across different states in each country, using the long-term (1985-2015) urban extent data (i.e., GAIA) and the sigmoid growth model. Given that the growth rates of urban area are notably varying during different urbanization stages (i.e., indicated by ISA), we used the sigmoid function to characterize the allometric growth of urban areas (Eq. 1 and Fig. S2.). That is, within each state, the growth of ISA in 1km grid in the next iteration can be determined from its current ISA level and the calibrated sigmoid growth model. During each iteration, grids with different urbanization levels have diverse ISA increments, resulting in spatially explicit differences in urban sprawl in the form of urban fractions.

$$ISA_t = a + \frac{b}{1+exp^{-c*(t-d)}} \tag{1}$$

where $ISA_t$ is the mean ISA value in built-up areas in a given region (e.g., state in our study) in year $t$; $a, b, c,$ and $d$ are four parameters that determine the sigmoid growth curve. Specifically, $a$ and $b$ represent the ISA level at the base level and the increment of ISA (e.g., amplitude) during the entire cycle of urban evolution, respectively; $c$ and $d$ are two parameters that jointly determine the shape of the sigmoid growth curve.

We incorporated the ISA-based growth mechanism with the Logistic-Trend-CA model (He et al., 2023), which incorporates temporal contexts of urban sprawl into the neighborhood configuration. The Logistic-Trend-CA model was developed from

the traditional Logistic-CA model (Hu and Lo, 2007; Wu, 2002), including the suitability surface (also known as transition rules), the neighborhood configuration, the stochastic perturbation, and the land constraint (see Supplementary texts). Compared to the Logistic-CA model, the adopted Logistic-Trend-CA model improved the neighborhood by using a trend-adjusted scheme (Eq. 2-3), referring to that newly developed urban neighbor pixels are more attributable to urban development than those urbanized neighbor pixels in earlier years (Li et al., 2020).

$$W_{ij}^{ts} = 1 - \frac{N_{ij}^u}{N} \tag{2}$$

$$\Omega_{ij}^t = \frac{\sum_{m^2} con(L_{ij}=developed) \times W_{ij}^{ts}}{m*m-1} \tag{3}$$

where $\Omega$ represents the neighborhood that considers the temporal contexts of urban sprawl using a weighting factor of $W_{ij}^{ts}$. $N_{ij}^u$ is the accumulated year of the cell $(i, j)$ with the status as urban from the annual urban time series data with a temporal interval of $N$, $m$ is the window size, and *Con()* is a conditional function, which returns 1 when the status of the cell $(i, j)$ is urban.

We obtained the urban development probability at the pixel scale from Eq. 4. Thus, during each iteration (i.e., 5-year interval), all grids were sorted in descending order according to their development probabilities, and those ranking ahead were preferentially considered in the modeling. The increment of the urban fraction was estimated for each pixel according to the calibrated sigmoid growth model (Eq. 2). We iteratively updated urban fractions in these pixels by estimating gradual ISA increments per iteration, until the total urban demand in each state has been allocated.

$$P_{dev} = P_{suit} \times \Omega \times Land \times SP \tag{4}$$

where $P_{dev}$ indicates the development probability; $P_{suit}$, $\Omega$, $Land$, and SP represent the suitability surface, neighborhood, land constraint, and stochastic disturbance, respectively. Details of these parameters can be referred to in the Supplementary texts.

**3.2 Calibration and validation of the Logistic-Trend-ISA-CA model**

We calibrated the Logistic-Trend-ISA-CA model at the state level using historical urban extent time series data (i.e., GAIA) from satellite observations (1985-2005). We evaluated the performance of derived global suitability using the receiver operating characteristic (ROC) approach, which essentially is a threshold-based evaluation approach (Sunde et al., 2014). That

is, the continuous values can be divided into binary maps using different thresholds to measure the agreement between threshold-derived results and the referenced urban extent (i.e., identified by their increased ISA during 1985-2005 with a threshold of 0.5) (Sunde et al., 2014). In this way, the area under the curve (AUC) is commonly used to quantitatively evaluate the performance of derived global suitability (Hosmer et al., 2013). It is worthy to note that here we used the traditional ROC approach to evaluate the suitability surface, which is only one component of the adopted urban CA model in this study, despite that our modeling target is ISA instead of binary urban extent. Additionally, we evaluated its performance of the ISA-based sigmoid growth model using the coefficient of determination ($R^2$) between the estimated and observed ISAs (1985-2015) at the state level.

Furthermore, we validated the model based on the root mean square error (RMSE) (Eq. 5) and coefficient of determination ($R^2$) between the modeled and observed ISAs (2005-2015) at the global scale. That is, we modeled the urban sprawl from 2005 to 2015 using the calibrated Logistic-Trend-ISA-CA model. In gneral, the relatively low RMSE and high $R^2$ suggest the calibrated urban CA model can capture the urban sprawl well.

$$RMSE = \sqrt{\frac{1}{n}\sum_{i=1}^{n}(ISA_{mod} - ISA_{obs})^2} \tag{5}$$

where $ISA_{mod}$ and $ISA_{obs}$ are the modeled and observed ISA results, respectively; $n$ is the pixel number.

**3.3 Projection of future global urban fractional change under eight scenarios**

To mitigate the uncertainty of future urban demand, we harmonized country-specific future urban area growth (2015-2100) from LUH2 with the urban area in 2015 obtained from satellite observations (i.e., GAIA). The harmonization process is inevitable because there is a distinct gap in urban areas in 2015 across different regions between these two datasets. Given that the GAIA data were derived from satellite observations with good quality and fine resolution, we harmonized future urban growth trends (2015-2100) from LUH2 under different SSP-RCP scenarios with the derived urban areas from GAIA in 2015. The harmonization of urban areas between these two datasets can be formulated as Eqs. (6-7).

$$Area_{harm,c,t} = Area_{LUH2,c,t} * \gamma \tag{6}$$

$$\gamma = \frac{Area_{obs,c,2015}}{Area_{LUH2,c,2015}} \tag{7}$$

where $Area_{harm,c,t}$ is the harmonized urban area of country $c$ in the year $t$; $Area_{LUH2,c,t}$ is the original urban area of country $c$ derived from LUH2 in year $t$; $\gamma$ is the harmonized rate between LUH2 and observed data (i.e., $Area_{obs,c,2015}$ and $Area_{LUH2,c,2015}$ are urban areas of the country $c$ in 2015, from satellite observations and LUH2, respectively).

We modeled future global urban fractional changes at 1km resolution under eight SSP-RCP scenarios. Here we assumed the trend of urban sprawl at the state level is consistent with that at the country level, as population and GDP change are commonly estimated at the country and regional scale (Doelman et al., 2018). Thus, we implemented the Logistic-Trend-ISA-CA model with the harmonized urban area growth from LUH2 and GAIA. Given that the conversion from non-urban to urban is commonly assumed to be irreversible (Li et al., 2015), we assumed that the urban area in regions with projected population decline in the future would plateau after reaching its peak.. Also, we compared our projected results with similar products regarding the provided details of gradual urban fractional change under eight SSP-RCP scenarios in a spatially explicit manner.

### 3.4 Uncertainty analysis

We calculated the delta harmonization rate, i.e., the quartile coefficient of dispersion (QCD) (Eq. 8) of harmonized rate $\gamma$ in each country during the overlap period of LUH2 and GAIA datasets. Due to the difference of adopted baseline urban extent in each product, there is a distinct gap regarding the urban area in these two datasets (i.e., GAIA and LUH2). Specifically, the urban extents in LUH2 were initially estimated from spatially explicit built-up data of Data and Information System Global Land Cover (DISCover) dataset at 1km resolution, which was mainly derived from the Advanced Very High Resolution Radiometer (AVHRR) satellite observations (Loveland et al., 2000; Goldewijk, 2017). While the definitions of "urban" are similar in both products, the differences in urban areas across various regions can be attributed mainly to their spatial resolutions and mapping years. In general, the urban extent in GAIA derived from Landsat has a longer temporal span and a high accuracy, with mean overall accuracies of above 90% across different years (Gong et al., 2020a). Moreover, the harmonized rate $\gamma$ can be different across countries and years. We calculated the QCD using the first (Q1) and third (Q3) quartiles for the annual harmonized rate $\gamma$ in each country during the overlap period, as Eq.8. The QCD can capture the

variation of harmonized rates within each country and is also comparable between countries with different deviation of future projection.

$$QCD = \frac{Q_3 - Q_1}{Q_3 + Q_1} \tag{8}$$

where $Q_1$ and $Q_3$ are the first (25th percentile) and third (75th percentile) quantiles of the annual harmonized rate in each country, respectively.

## 4. Results and discussion

### 4.1 Future global urban fractional change under eight scenarios

There are distinct spatial variations across countries regarding urban area growth rates during 2015-2100 under eight SSP-RCP scenarios (Fig. 2). Among these scenarios, Global South countries located in middle Asia, south America, and Africa, would likely experience more noticeable urban growth than Global North countries in the future (Fig. 2), i.e., the average growth rate (i.e., 2100/2015) of Global South countries is around 5 under various future scenarios, while the average growth rate in Global North countries is mostly lower than 3. It is worth to note that the future urban growth in China is relatively low in these scenarios, mainly due to the nearly plateaued population growth by 2030, which is distinctly different from the continuously increasing population in other developing countries such as India and Nigeria (Chen et al., 2020b). In addition, the urban growth rates in the USA are relatively low among different SSP-RCP pathways, except for SSP5 with Fossil-fueled development (Fig. 2) (O'neill et al., 2017), suggesting that future socioeconomic development may significantly impact the urban area growth in different regions and countries.

The country-specific urban sprawl is notably different under eight SSP-RCP scenarios in the future (Fig. 3). Considering different urban development pathways from 2015 to 2100, we selected three specific countries (i.e., the USA, China, and Nigeria) for illustration. Spatially explicit patterns of urban sprawl were compared under three kinds of scenarios: low-, median-, and high- growth, which were selected from eight SSP-RCP scenarios regarding the total urban area growth in each country (Fig S4). To be specific, the urban growth of the USA is highest in SSP5 and lowest in SSP4 (Fig. 3b), which is different from Nigeria (Fig. 3a) and China (Fig. 3c). Apparently, under the high-growth scenario, more natural lands would be

developed as urban, resulting in a discernible urban sprawl with connected pixels around urban centers (Fig. 3, a-c). Besides, it is worth noting that China's urban area will continue to grow until around 2050, although the peak of the population is likely to plateau by around 2030, whereas the per capita urban area is anticipated to keep growing (Fig. 3c) (Li et al., 2019b). Our result can also reflect the gradual urban fractional change across years throughout 2100 (Fig. 4). For instance, small human settlements in Nigeria are likely to grow in the future, and most of them occur in small settlements (Fig. 4a). In contrast, in the USA, most newly developed urban areas in the future are likely to occur in/around the city center (Fig. 4b). The urban fractions both in the urban and rural areas would increase and plateau around 2050 in China (Fig. 4c), showing different temporal trends as reflected by the USA and Nigeria. In addition, the spatially explicit pattern of urban sprawl in our dataset is consistent with historical observations (GAIA), as illustrated in China (Fig. 5), showing the relatively complete urban evolution from early 1985 to 2100. Similar long-term dynamics of urban sprawl in the form of ISA can be found in Fig. S5-6. In addition to the suitability, the state-based trend of ISA growth from satellite time series data may also impact the ISA growth at the pixels, particularly for those with extremely low and high ISA values. It's worth noting that the ISA-based growth in our modeling mechanism may underestimate the growth of pixels with very low ISA values or non-developed, although the stochastic disturbance term has been involved in our modeling mechanism. Meanwhile, the rate of urban fractional growth is slow for pixels around the city centers with relatively high ISA values. Appropriate strategies by constraining the filling of urban inner spaces and the expansion of urban bound should be developed to improve the spatial allocation of urban CA model.

4.2 Model performance

4.2.1 Spatially explicit ISA growth model

The parameters revealed from the sigmoid model show noticeable spatial variations across different states worldwide (Fig. 6). In total, there are four parameters in characterizing the urban area growth over the past decades using Eq. (1), including the initial urbanization level ($\alpha$), the increment of ISA during the growth ($b$) (i.e., amplitude), and the rate of urban growth ($c$) during the most rapid growth period ($d$). The parameters $\alpha$, $b$, and $c$ indicate the corresponding ISA rate (0-1), while parameter $d$ represents the most urbanized year. The spatial distribution of urban growth patterns revealed from the sigmoid model is

probably caused by varying ISA growth patterns across spaces. The initial urbanization level ($\alpha$) is steadily around zero, as illustrated in Fig. 6a, except for those states in North America and Asia with low urbanization levels (e.g., without distinct growth of urban areas over past decades). The increment of ISA ($b$) follows the general pattern of urban evolution in the urban cycle of each state, agreeing well with the common urbanization level from 60% to 90%, especially in countries that experienced a fast urbanization process over the past decades (e.g., China) (Fig. 6b). Besides, the rate of urban growth ($c$) (Fig. 6c) changes steadily from 0.1 to 0.25 and the corresponding year with the fastest growth rates ($d$) (Fig. 6d) mainly falls into the range 1995-2015, which jointly determine the shapes of state-specific sigmoid curves in the modeling process. Such a spatially explicit ISA growth pattern suggests diverse urban growth stages over the past decades, mainly gained from the long-term annual urban extent time series data (GAIA). The spatially explicit urban area growth patterns with parameters can well reflect the pathway of urban development in regions with different urbanization levels. The fit performance (i.e., with $R^2$ above 0.8) of the sigmoid model at the state-level indicates the calibrated model can well characterize the spatially explicit urban growth patterns over past decades using satellite observations (Fig. 7). It is worthy to note that there are some states with relatively low performance, probably due to limited increments with different urbanization stages, e.g., in highly urbanized regions or developing regions with low urbanization levels.

4.2.2 Performance of the suitability surface

The derived suitability surface from the Logistic regression model can well separate the urbanized and persistent regions (Fig. 8). Countries in East Asia (e.g., China, Mongolia, Thailand), West Europe (e.g., France, Germany), and North America (e.g., USA) have better performance than other regions, with AUC values greater than 0.8 (Fig. 8b). For instance, the AUC in China is above 0.9, suggesting the developed model can well distinguish those urbanized regions from the spatial proxies, as China experienced an unprecedented urban expansion over past decades (Gong et al., 2020a). However, the model performances in Canada, Afghanistan, and West African regions (e.g., Zambia), are relatively worse compared to other regions, probably due to the poor quality of these spatial proxies (Fig. 8a) or relatively small urban growth (Fig. 2), i.e., the ROC curves in Sudan

and Zambia are associated with lower AUCs, different from those in China and USA (Fig. S7). Furthermore, the suitability surface performs well in most states with relatively high AUC values (Fig. S8).

4.2.3 Performance of Logistic-Trend-ISA-CA model

The proposed Logistic-Trend-ISA-CA model can achieve a good performance at the global scale, with the overall R2 of 0.9 and the RMSE of 0.08 (Fig. 9). Overall, over 30% of countries have relatively low ISA differences between observed and modeled results, i.e., with ISA difference ranging from -0.01 to 0.01 (Fig. 9a). Only around 3% of global countries are associated with considerable over- or under-estimations (Fig. 9a). The modeled ISAs in Global South countries (e.g., India and Bolivia) are slightly overestimated compared to satellite observations; whereas in northern regions such as Ukraine and Uzbekistan, our modeled results are relatively underestimated. Although the under- and over-estimations of modeled ISAs in our results are not so evident, the reasons behind them are likely related to the relatively low suitability surface (Fig. S1-f) or the low rates of urban sprawl in these regions over the past decades (Gong et al., 2020b). For example, the modeled ISAs in Global South countries are slightly underestimated in general due to the initial urbanization stages with relatively slow growth rates over the past decades in these regions (Fig. 9a). Also, patterns of suitability surface (Fig. S1-f) and ISA difference are not always consistent across regions. Although the performance of the suitability surface is relatively worse in Canada (Fig. 8a), the modeled ISA difference is small. Moreover, the underestimated ISA during the modeling process was considerably reduced in our results (Fig. 9b), because those overestimated errors in the traditional urban CA model can be mitigated due to the fractional increase mechanism in our model, especially in regions (e.g., China) with massive discrete urban landscape.

4.2.4 Harmonized urban demands

The adopted scheme of urban demand harmonization considerably mitigated the gap in urban areas across different countries between LUH2 and satellite observations (i.e., GAIA), which was regarded as a reference in our study. Countries with urban harmonized rates $\gamma$ less than 1 (i.e., green color in Fig. 10a) represent that the urban demand derived from LUH2 are overestimated, especially for those Global-South countries in central Africa and South America, due to the relatively coarse resolution global urban extent product (i.e., the History of the Global Environment database, HYDE). Similarly, countries with

harmonized rates $\gamma$ above 1 (i.e., orange color in Fig. 10a) are likely to be underestimated in LUH2 regarding the urban

demands, such as China, Canada, the USA, Russia, and most European countries. The uncertainties caused by LUH2 can be

reduced by using satellite-derived urban extent time series data (i.e., GAIA), which is equipped with fine spatial resolution and

good data quality. Overall, our harmonized urban area results follow the general trend of historical urban development

meanwhile mitigate the uncertainties of country-specific urban areas in LUH2. Besides, the harmonized future urban demands

at the global scale vary across different SSPs (Fig. 10b). Urban sprawl under SSP2 (middle of the road) and SSP3 (regional

rivalry) is notably slower than in the observed period (1985-2015), while the urban sprawl under SSP5 (fossil-fueled

development) is much higher (Fig. 10b). Specifically, urban expansion under SSP3-RCP 7.0 has lowest urban demands in the

future. In contrast, the urban growth is highest in SSP5-RCP8.5 (Fig. 10c), with a relatively large area difference across

different countries. Our results indicate the QCD is relatively small (0-0.2) in most countries, suggesting the $\gamma$ is robust during

the overlap period (Fig. S9). The urban areas in most regions like America, Canada, China, India, and Australia have similar

trends in GAIA and LUH2, while their gaps (i.e., QCD) are relatively large in those least developed countries (Fig S9). It is

worthy to note that here we directly inherited the future trend of urban areas from the integrated assessment model (IAM)

under diverse SSP-RCP scenarios (Hurtt et al., 2011) across different states in each country, harmonized with historical urban

extent dynamics from satellite observations. However, the urbanization stage was not considered in those IAM models, which

were mainly driven by demographical and socioeconomic factors. In the future, the urbanization stages could be a weight

factor when downscaling urban areas from country to state.

4.3 Data comparison with similar global urban extent products

Our results can provide spatially explicit information of urban fraction compared to other global urban sprawl products, such

as Gao and O'neill (2020) and Chen et al. (2020a). It is worth noting that the spatial patterns were compared with other global

urban products under the SSP2 scenario (i.e., middle of the road), under which scenario the urban area growth follows the

historical trend in general (Fig. 11). Overall, our modeled results can provide more detailed information of urban fraction at

1km spatial resolution around the urban core and rural areas (Fig. 11). In Gao and O'neill (2020) 's result, due to the relatively

coarse spatial resolution, many details of the urban extent and the urban fractional change have been ignored (Fig. 11). In Chen et al. (2020) 's result, the urban pixels in/around the urban fringe indeed are associated with relatively low urban fractions compared to those in the urban core (Fig. 11). Supported by the long-term urban extent time series data and the ISA-based modeling scheme, our results can maintain many details of urban intensity (i.e., both high and low intensity levels) from the city core to surrounding rural areas in a spatially explicit manner. Continuous urban fractional change at a fine resolution can be better applied in global urban studies with notably improved spatial details and reduced uncertainties. The overall trends of future global urban sprawl in our results are similar as those two global urban extent datasets (Chen et al. (2020a); Gao and O'neill (2020)), but their magnitudes are notably different (Fig. S10). These deviations are mainly attributed to the variation caused by the urban area growth estimation model adopted in different products. For instance, the temporal trend of our results was mainly inherited from LUH2, which essentially was estimated from multiple integrated assessment models. However, for products in Chen et al. (2020) and Gao et al. (2020), their urban areas were estimated using panel analysis and data-driven approaches, respectively, based on four-epoch time series data of Global Human Settlement Layer (GHSL).

Our dataset can model spatially explicit urban fractional change at a 1km spatial resolution and can well capture the different intensity of urban development (e.g., New York City in the USA) under eight scenarios (Fig.12). Under different scenarios in the future, which were selected from different potential magnitudes of urban growth (Fig. S4), the spatial patterns of urban sprawl in New York City under different scenarios show more spatial details and continuous urban fractional change in our results than that revealed in Gao and O'neill (2020) and Chen et al. (2020a) (Fig.12). Under the low urbanization pathway (SSP4 derived from Fig S4), urban sprawl was under-estimated around/in urban fringe in our result and Chen et al. (2020a), while Gao and O'neill (2020) simulated more intensive urban sprawl in urban fringe under this scenario (Fig.12b). Under the high urbanization pathway (i.e., SSP5 derived from Fig S4), there is some over-estimated urban land in urban fringe in Gao and O'neill (2020) and Chen et al. (2020a), however, less over-estimation was occurred around urban fringe in our result (Fig.12c). This is probably due to the mechanism of sigmoid growth model we selected that correspond with historical urban development in this region, which differs from Gao and O'neill (2020) and Chen et al. (2020a).

**5 Data availability**

The gridded dataset of global urban fractional change (2015-2100, 5-year interval) at 1km spatial resolution under eight future development pathways and the global urban development probability map can be viewed and downloaded from https://doi.org/10.6084/m9.figshare.20391117.v4 (He et al., 2022). The historical annual long-term urban extent data (30m resolution) were derived from annual Global Artificial Impervious Area (GAIA) data (http://data.ess.tsinghua.edu.cn/) (Gong et al., 2020a). Future urban area growth trends across different countries were derived from the Land Use Harmonization (LUH2) data (https://luh.umd.edu/data.shtml) (Hurtt et al., 2020).

**6 Conclusions**

In this study, we developed a gridded dataset of global urban fractional change (2015-2100, 5-year interval) at a 1km spatial resolution, under eight scenarios of socioeconomic pathways and climate change. We first characterized ISA growth patterns and developed a state-specific ISA-based growth model based on long-term observations, using the sigmoid model to account for different urban growth rates under varying urbanization levels. Then, through incorporating the ISA-based growth mechanism with the CA model, we calibrated the state-specific urban CA model with evaluations at the global scale quantitatively. Finally, we projected future urban gradual changes at 1km resolution under eight future development pathways with harmonized urban growth demand from the GAIA and LUH2 datasets.

Our database can provide temporally consistent and spatially explicit urban fractional changes under eight development pathways. It is worth noting that the temporal contexts of urban evolution were comprehensively considered in our projected dataset, using long-term and annual urban extent time series data (i.e., GAIA). The average urban growth rates derived from the LUH2 dataset at the country level show significant differences under eight future SSP-RCP scenarios, of which Global South countries/regions are primary drivers of future global urbanization. In addition, the spatially explicit ISA growth patterns differ across states, reducing the uncertainties in urban sprawl modeling. Besides, the urban demand harmonization process considerably mitigated the gap of projected urban areas across different countries between LUH2 and GAIA. Furthermore, the overall $R^2$ and RMSE of modeled and observed ISAs are 0.9 and 0.08, respectively, suggesting an overall good performance

of urban sprawl modeling. Compared to other global urban products under future scenarios, our results can promote future urban land use efficiency by simulating gradual urban fractional change with notably improved spatial details (i.e., 1km) (Chen et al., 2020a; Gao and O'neill, 2020; Li et al., 2019a; Li et al., 2021).

The global dataset of gridded urban fractional changes under eight SSP-RCP scenarios is of great potential to support various global urban studies. For example, future urban extents with fractional information can delineate the intensity gradient from the urban core to rural areas, which is helpful for relevant studies such as urban heat island estimation and human-natural interactions in urban ecosystems (Acuto et al., 2018; Castán Broto and Bulkeley, 2013; Klein Goldewijk et al., 2010). Besides, our developed dataset can serve as a base input for global integrated assessments (e.g., urban energy consumption, inequality evaluation) (Zhou et al., 2022) and Earth system models, in which the anthropogenic activities within the urban extent can be quantitatively measured (Li et al., 2014; Li and Gong, 2016).

**Author contributions**

LX and ZY designed the research; HW and LX implemented the research and wrote the paper; ZY, SZ, YG, HT, WY, HJ, BT., SZ, LX, and GP revised the manuscript.

**Competing interests**

The authors have the following competing interests: Xuecao Li and Yuyu Zhou are two editors of Earth System Science Data.

**Acknowledgements**

This work was funded by the National Natural Science Foundation of China (42101418), the NSFC Excellent Young Scientists Fund (Overseas), and the Chinese University Scientific Fund.

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

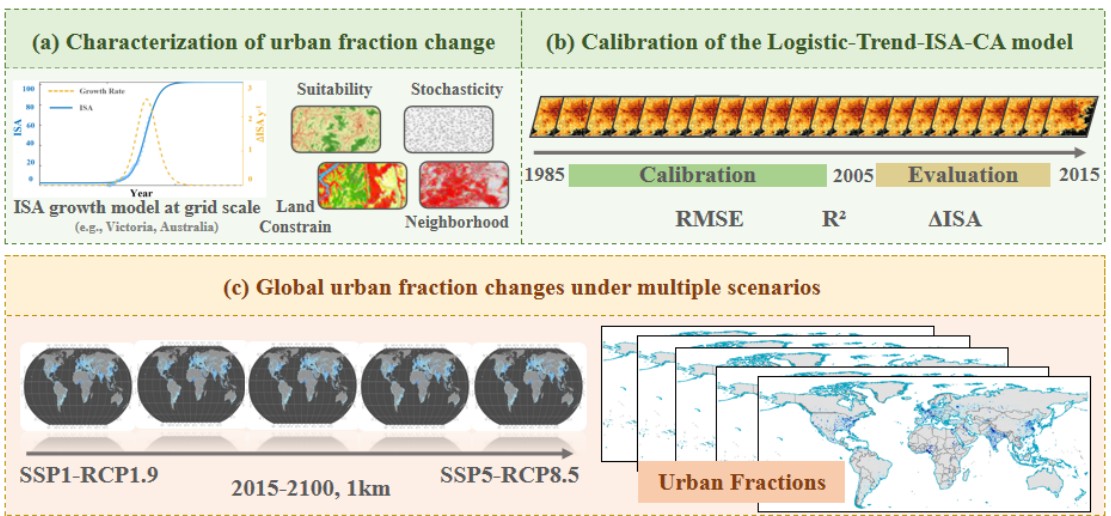

**Fig. 1**: The proposed overall framework of global urban fraction dynamics under future scenarios using the developed ISA-based urban CA model, including the characterization of urban fraction change with urban CA model (a), the calibration of Logistic-Trend-CA model (b), and the projection of future global urban fraction changes under 8 scenarios (c).

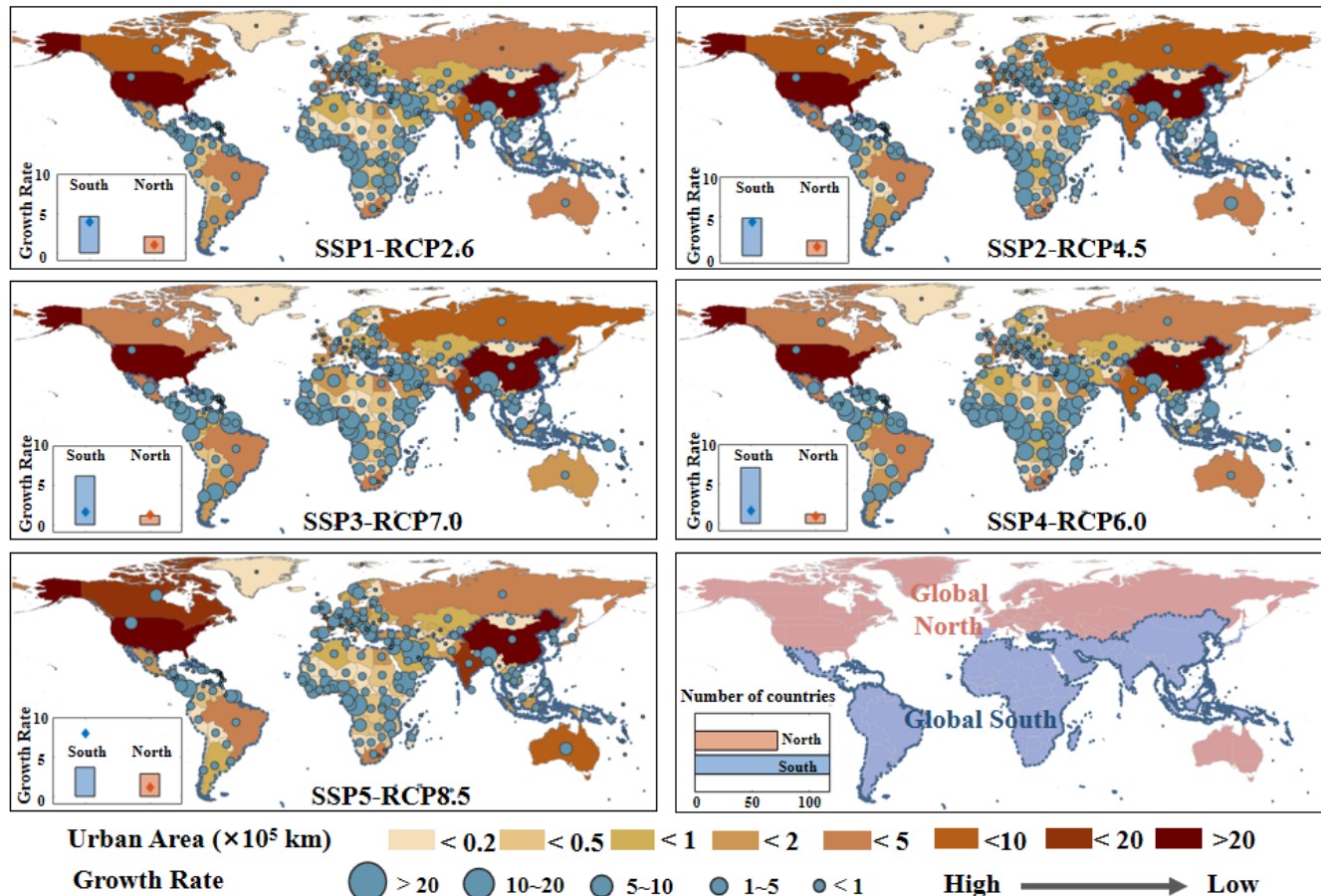

**Fig. 2**: The average urban growth rate (2100/2015) derived from LUH2 at the country level and the comparison between Global South and North countries, under 8 future development pathways. The USA and China are represented as solid dots in boxplots, as representative countries in Global South and North countries. Detailed comparison among 8 RCPs in a common SSP scenario can be found in Fig. S3.

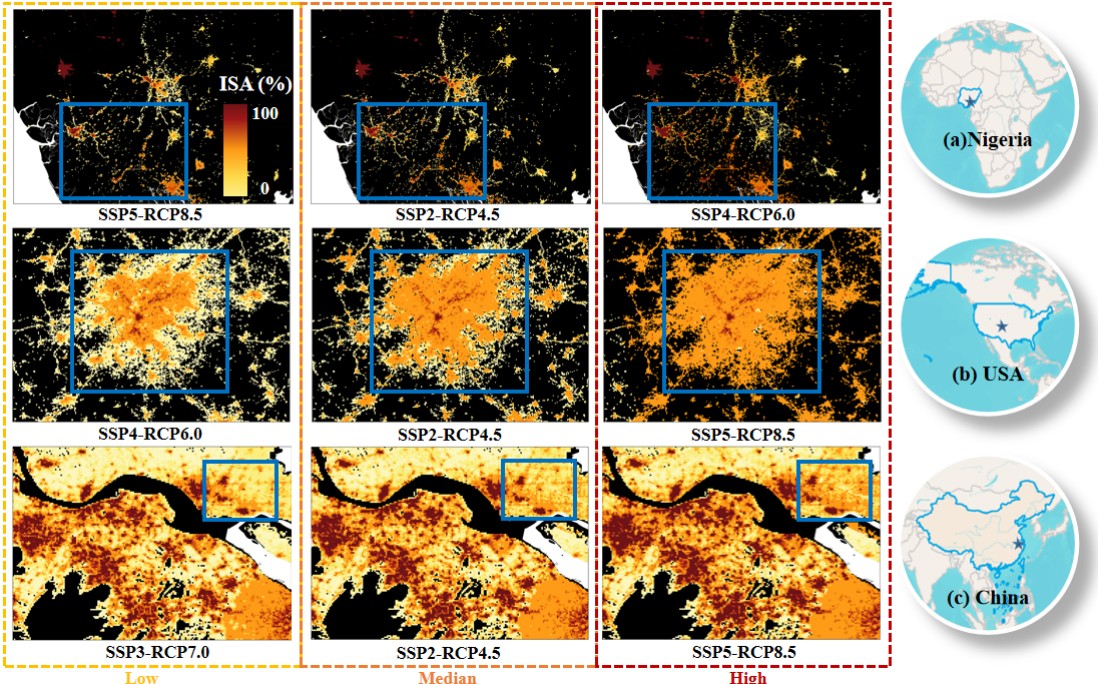

**Fig. 3**: The modeled urban sprawl in 2100 with urban demand change in Nigeria (Enugu) (a), the USA (Atlanta) (b), and China (Yangtze River Delta) (c) under typical SSP-RCP scenarios selected by grouping future urban demand in Fig S4.

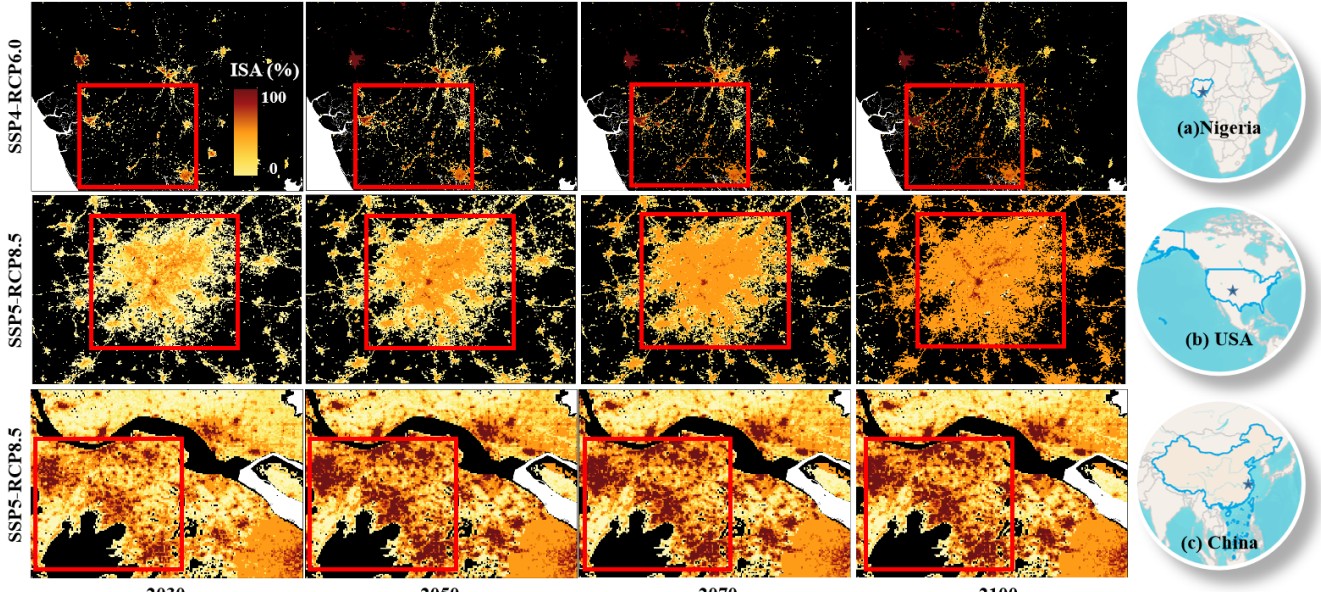

**Fig. 4**: The projected spatial patterns of urban sprawl in some typical urbanization regions of Nigeria (a), the USA (b), and China (c) from 2030 to 2100 under the most fluctuating scenario.

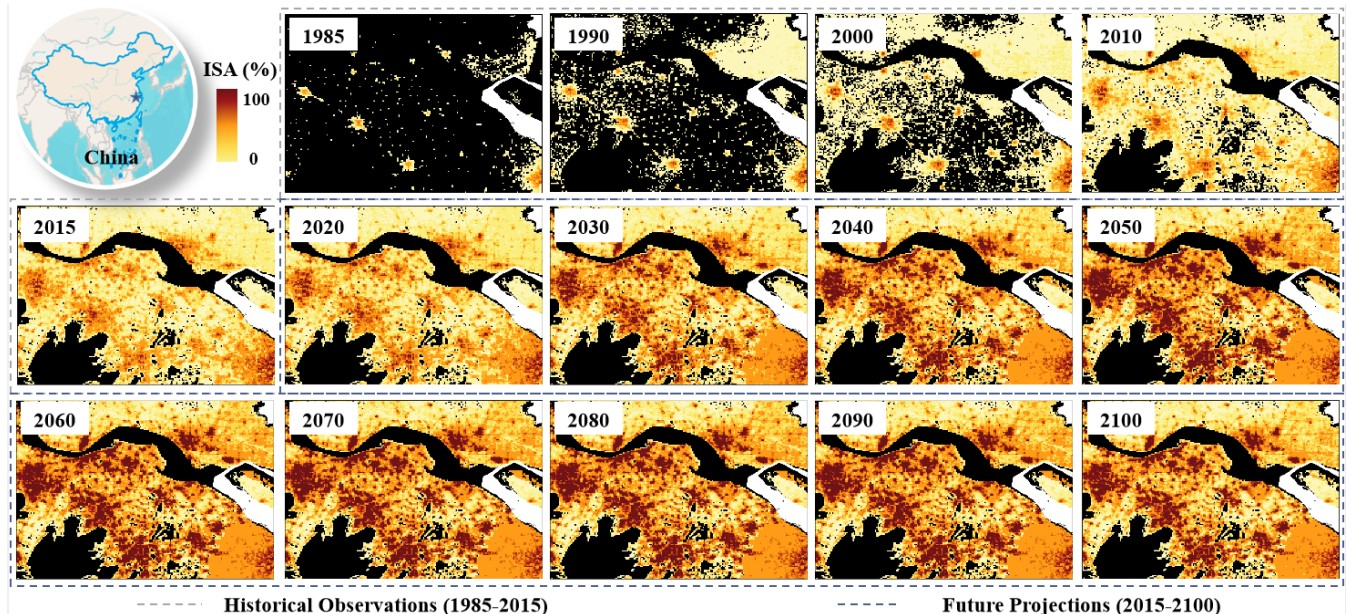

**Fig. 5**: Change of urban sprawl in China at 1km spatial resolution from 1985 to 2100 under the SSP5-RCP8.5 scenario.

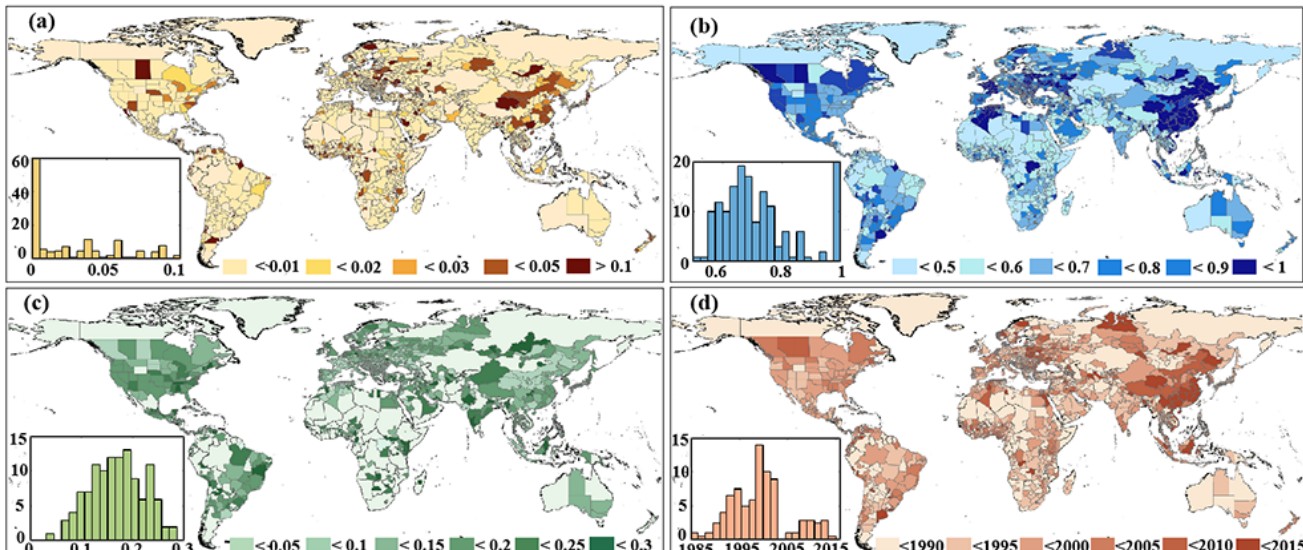

**Fig. 6**: The spatial distribution of the initial urbanization level (a), increment of ISA during the growth (b), the fastest growth rate (c) during the most rapid growth period (d) of the derived sigmoid growth curve in each state.

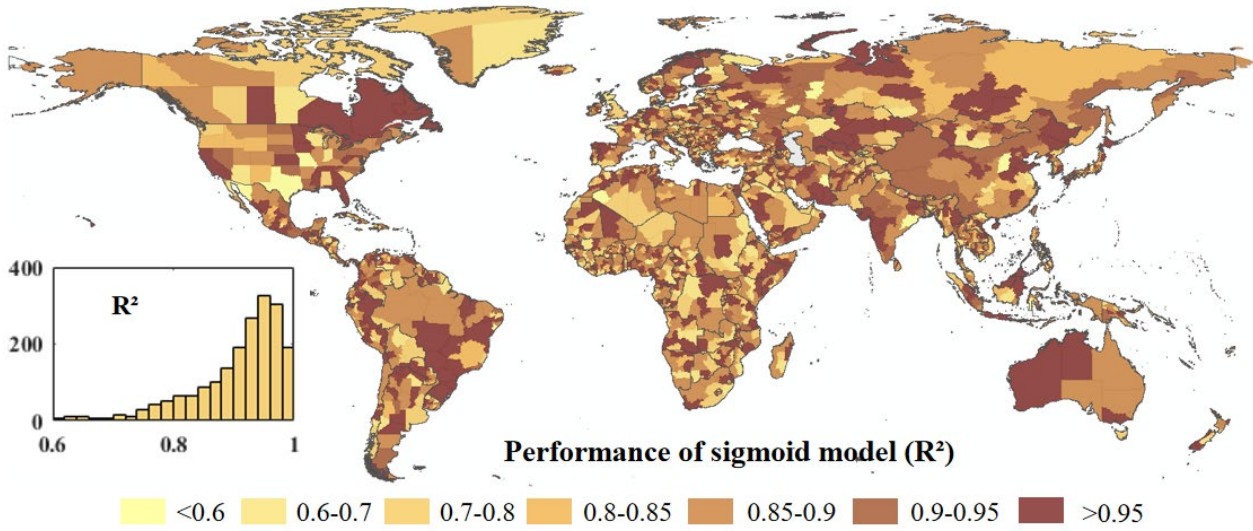

**Fig. 7:** The fit performance (R²) of state-level sigmoid model at the global scale. R² is the coefficient of determination between the simulated and referred ISAs over past decades.

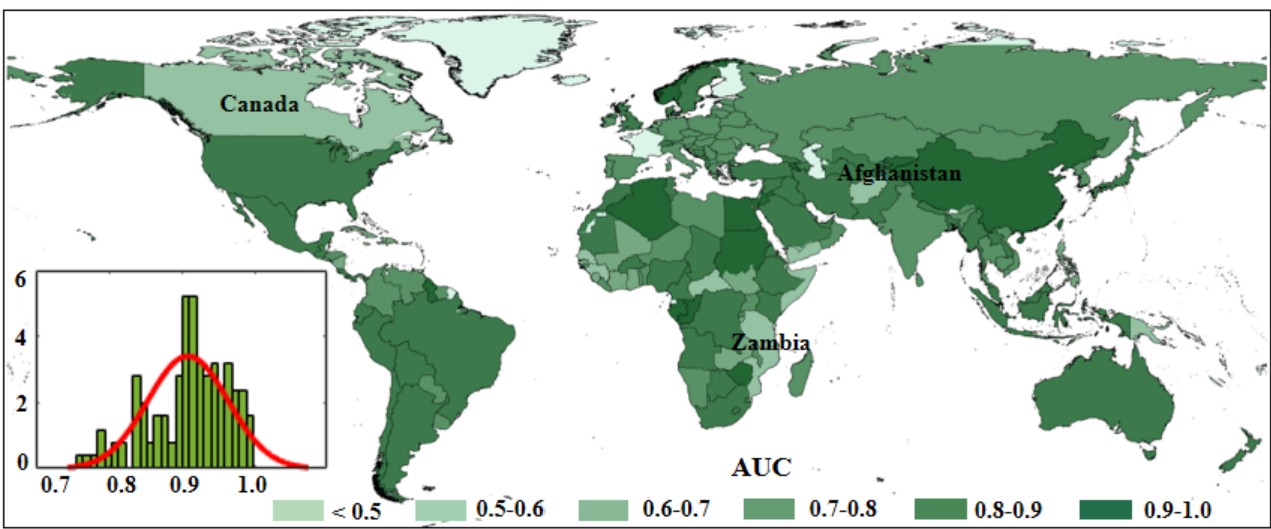

**Fig. 8**: The model performance of derived suitability surfaces using the indicator of the area under the curve (AUC) at the global scale. The ROC curves in some representative countries can be found in Fig S7.

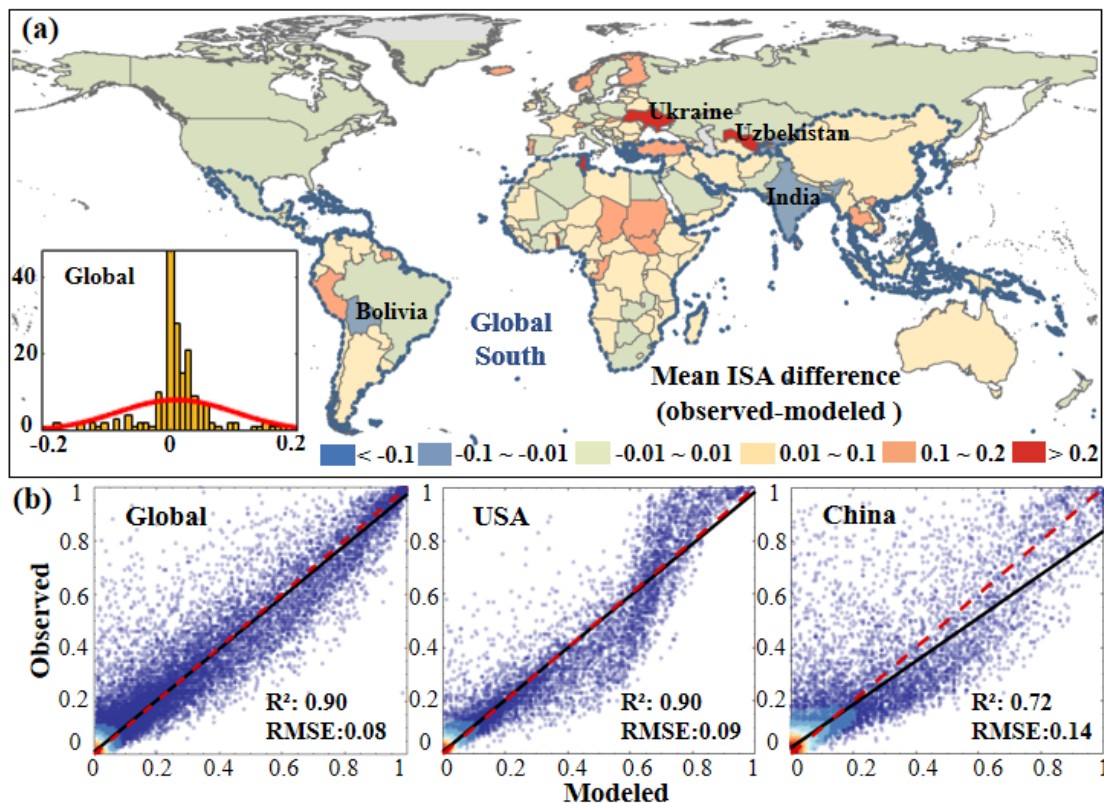

**Fig. 9**: The spatial distribution of the ISA difference between modeled and observed results in 2015 (a) and the scatter plots (b) of these two datasets at the global scale and typical countries (i.e., USA, China).

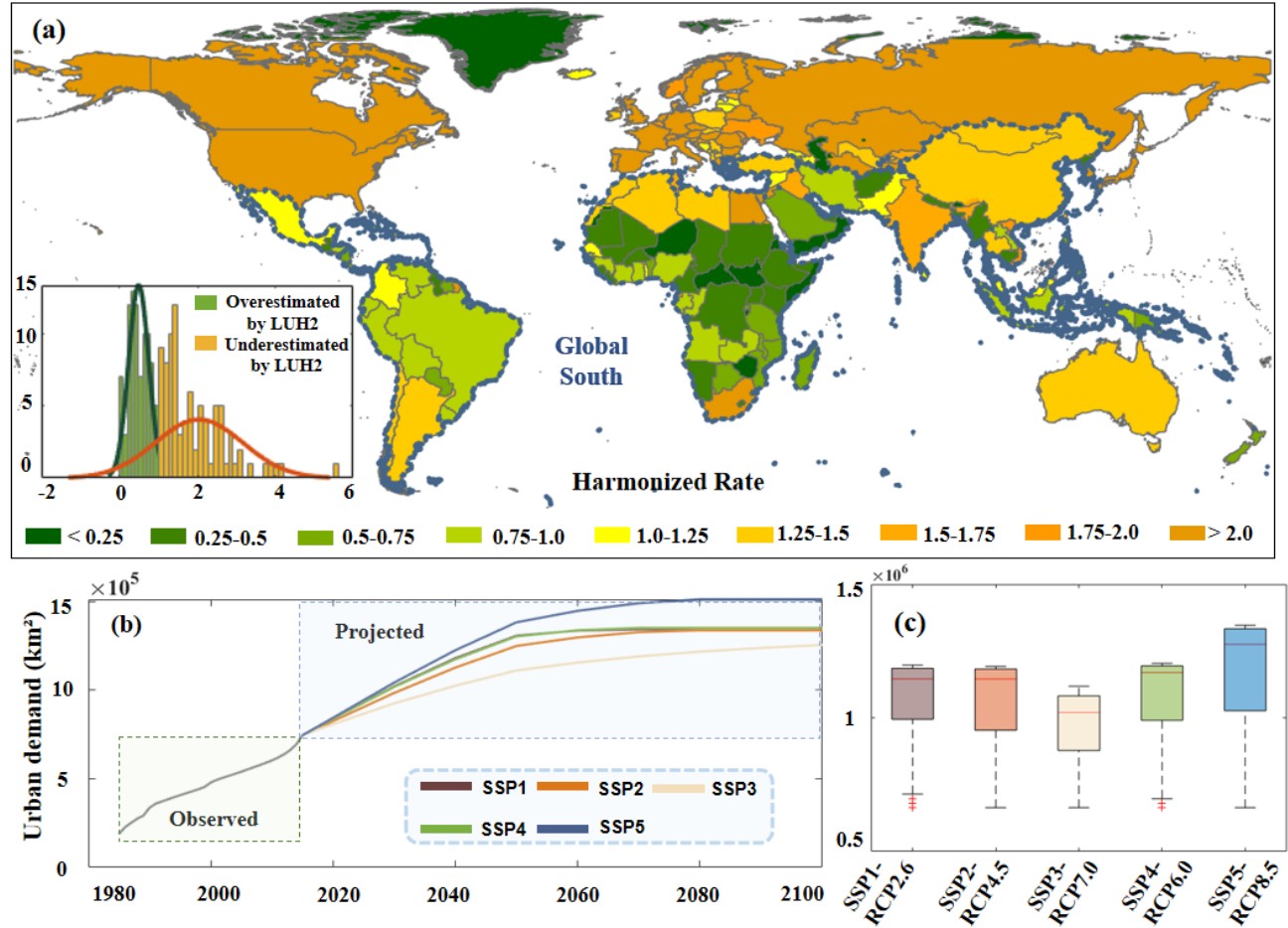

**Fig. 10**: The spatial distribution of harmonized rates at the country level in the base year 2015 (a), trends of global urban demand after harmonization (b), and the comparison across 8 scenarios (c).

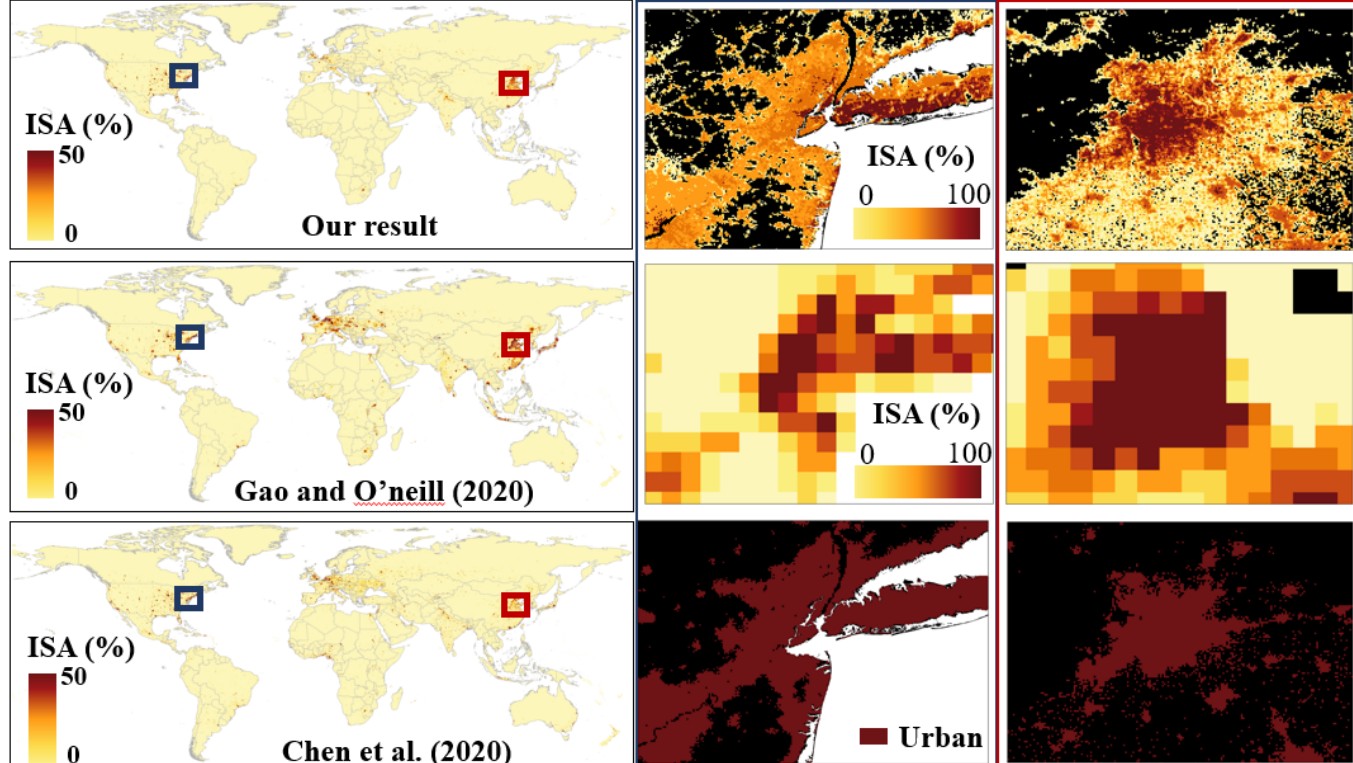

**Fig. 11:** Comparisons of our results with two similar projection products from Gao and O'neill (2020) and Chen et al. (2020a) under SSP2-RCP4.5 in 2100. An overview of urban sprawl at global scale and two typical cases in the USA and China of relatively high intensity regions.

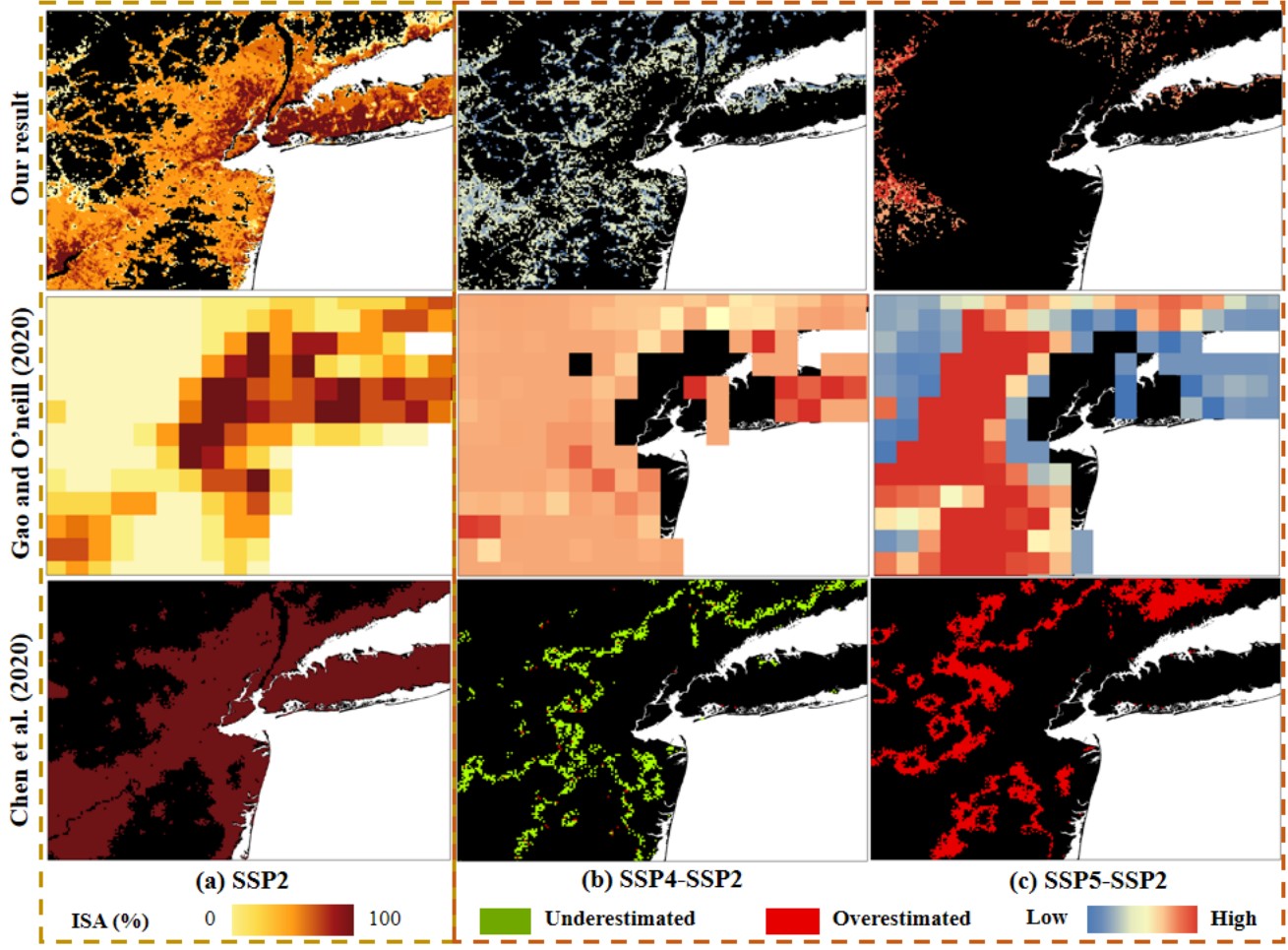

**Fig. 12:** Comparisons of modeled urban patterns in New York City (USA) with our results, Gao and O'neill (2020), and Chen et al. (2020a) under general (SSP2-RCP4.5) (a), low urbanization (SSP4-RCP6.0) (b), and high urbanization (SSP5-RCP8.5) (c) pathways in 2100. The modeled results under SSP2-RCP4.5 were considered as the base for comparison.

**Table 1**: The adopted spatial proxies in this study.

| Spatial proxies | Description | Source |
|---|---|---|
| Land | Land cover | Moderate Resolution Imaging Spectroradiometer (MODIS) Land Cover Dynamics (MCD12Q2) (https://doi.org/10.5067/MODIS/MCD12Q2.006) |
| | Protected area | The World Database on Protected Areas (WDPA) (http://wcmc.io/WDPA_Manua) |
| Location | Major cities | World city centers (http://ngcc.sbsm.gov.cn/article/zh/) |
| | Traffic | World major roads, highways, and local roads (https://www.openstreetmap.org) |
| Terrain | Elevation | Shuttle Radar Topography Mission - Digital Elevation Model (DEM) (http://earthexplorer.usgs.gov/) |
| | Slope | Derived from DEM |