# Peer review of "Global urban fractional changes at a 1km resolution throughout 2100 under eight SSP-RCP scenarios"

_Earth System Science Data, 2022_

## Author Comment (AC1)

**Reviewer1**

**Comment #1-0**

This paper presents a new global 1km fractional urban change dataset for the 2015-2100 period. It is the first global fractional urban change dataset that I am aware of, which should make it of high interest to the Earth and Environmental Science community. The methodology used was adapted from previous methods developed by the authors to allow for fractional (rather than binary) urban change modelling. The model validation results (e.g., RMSE = 0.08) are encouraging, although I request the authors to explain the calibration/validation procedure in more detail (see my Specific comments 4-5). The manuscript is generally well structured and readable, but could use a language and grammar check. Overall, I believe this paper is a valuable addition to the scientific literature on urban change, and that it could be publishable after further revisions and clarifications by the authors.

**Response:** thank you for the positive comments and valuable suggestions. As suggested, we have carefully revised the methodology part (especially in the calibration/validation) and checked the grammar in our updated manuscript. The detailed point-by-point response is presented below.

**Comment #1-1**

Page 3, line 10: "Although several global datasets of urban extent dynamic with conversions from non-urban to urban have been proposed, there is still limited effort to characterize the gradual urban fractional change (i.e., ISA) within each grid when projecting future global urban sprawl (Potere et al., 2009; Huang et al., 2021; Herold et al., 2003; Seto et al., 2012; Li et al., 2017)". I suggest to include more information on some of these other global urban extent datasets, e.g., their spatial resolution, the data used to calibrate/validate the model, the years for which the data is available (e.g., to 2050? 2100?). Also, you may want to note if any are not freely available for download. This additional information can help to highlight the other advantages of your dataset (aside from its mapping of fractional cover).

**Response:** thank you for your suggestions and questions. As suggested, we included information of other global urban extent modeling work in our revised manuscript (*Chen et al., 2020*; *Gao and O'neill, 2020*; *Zhou et al., 2019*). In general, the spatial resolution of these products is either relatively coarse (e.g., 8km; *Gao and O'neill, 2020*) or only available in binary format (1km; *Chen et al., 2020*; *Zhou et al., 2019*). Moreover, limited attempts have been made to fully explore the temporal dynamics of urban evolution from long-term and annual urban extent time series data (*Chen et al., 2020*; *Gao and O'neill, 2020*), which could introduce noticeable uncertainties in projecting future urban growth. Yes, these datasets are free to download. We improved our descriptions in our revised manuscript as below.

*"However, the spatial resolution of these global urban products is either relatively coarse (8km) (Gao and O'neill, 2020) or only available in binary format (1km) (Zhou et al., 2019; Chen et al., 2020a). The temporal contexts of urban sprawl were limitedly considered in these studies, leading to noticeable uncertainties regarding the projected global urban extent dynamics in the future with long-term historical urban sprawl." (Page 3, Line 13-16)*

**Comment #1-2**

Page 4, line 15: It would be beneficial to readers if you can explain why the global artificial impervious area (GAIA) dataset was used for this model calibration and validation. For example, are there no appropriate ~1km fractional urban cover maps that could have been used for this? My concern is that it GAIA a binary urban/non-urban map that was resampled to 1km, and not a "true" fractional cover map.

**Response:** thank you for your suggestion. We acknowledge that there is currently no such dataset of ISA time series information directly from remotely sensed observations. As such, generating ISA time series data through spatial aggregation from high-resolution urban extent data is a commonly adopted strategy. Moreover, it is worth to note that the GAIA data is of high quality with mean overall accuracies of above 90% across different years. Thus, the derived ISA maps from GAIA can well characterize the urban fractional information within each 1km grid. We explained this issue in our revised manuscript.

*"Given that there are currently no urban fractional ISA dataset in high spatial resolution (e.g., 1km) directly obtained from satellite observations, here we adopted the commonly used strategy through spatial aggregation from high-resolution (e.g., 30m) urban extent data to derive the ISA time series data for modeling. The GAIA data record annual global urban extent (i.e., non-urban and urban) at a 30m resolution, spanning from 1985 to 2018, with overall mean accuracies above 90%. Besides, the derived historical urban extents from GAIA are temporally consistent (i.e., non-urban to urban) over past decades." (Page 4, Line 16-21)*

**Comment #1-3**

Page 5, line 5. More information is needed on these spatial proxies, and how they were considered in the model. I suggest to add the references for each dataset used in Table 1, as well as how the spatial proxies were derived from these datasets (e.g., based on distance to the features like city centers/roads/protected areas/MODIS land cover types?).

**Response:** thank you for your suggestions. As suggested, we provided detailed information about these spatial proxies in our revised manuscript and the supplementary materials. These spatial proxies were mainly derived by calculating pixel-based distance to the nearest roads and locations, directly derivation from terrain (e.g., DEM and slopes), or specific constrains such as protection area. Details of these spatial proxies and their descriptions can be referred in Table 1.

*"For example, some spatial proxies (e.g., land cover and protected area) were defined as specific constrains (e.g., suitable or not), while terrain and location proxies were directly calculated from the DEM and the distance to the nearest roads (or city centers), respectively." (Page 5, Line 13-16)*

**Table 1**: The adopted spatial proxies in this study.

| Spatial proxies | Description | Source |
|---|---|---|
| Land | Land cover | Moderate Resolution Imaging Spectroradiometer (MODIS) Land Cover Dynamics (MCD12Q2) (https://doi.org/10.5067/MODIS/MCD12Q2.006) |

| | | |
|---|---|---|
| | Protected area | The World Database on Protected Areas (WDPA) (http://wcmc.io/WDPA_Manua) |
| Location | Major cities | World city centers (http://ngcc.sbsm.gov.cn/article/zh/) |
| | Traffic | World major roads, highways, and local roads (https://www.openstreetmap.org) |
| Terrain | Derived from dem | Shuttle Radar Topography Mission - Digital Elevation Model (DEM) (http://earthexplorer.usgs.gov/) |

**Comment #1-4**

Page 7, line 14: "That is, the continuous values can be divided into binary maps using different thresholds to measure the agreement between threshold-derived results and the referenced urban extent. In this way, the area under the curve (AUC) is commonly used to quantitatively evaluate the performance of derived global suitability (Hosmer et al., 2013)." Is this binary validation necessary, considering the purpose of the model was to generate fractional urban cover estimates? If so, I suggest to explain why.

**Response:** thank you for this question. Yes, we modeled urban fractional changes within each 1km grid, which is notably different with traditional modeling (i.e., from non-urban to urban directly). The suitability surface is one of the main components in the proposed ISA-based urban CA model, reflecting the biophysical priority for urban development. Therefore, we evaluated the performance of derived global suitability using the traditional ROC approach, which essentially is a threshold-based evaluation approach. That is, we identified those changed and persistent pixels using the threshold of 0.5 (i.e., $\Delta ISA$ during 1985-2005) and then evaluated the performance of suitability surface using the indicator of AUC (*He et al., 2023*). We explained this issue in our revised manuscript.

*"We evaluated the performance of derived global suitability using the receiver operating characteristic (ROC) approach, which essentially is a threshold-based evaluation (Sunde et al., 2014). That is, the continuous values can be divided into binary maps using different thresholds to measure the agreement between threshold-derived results and the referenced urban extent (i.e., identified by their increased ISA during 1985-2005 with a threshold of 0.5). In this way, the area under the curve (AUC) is used to quantitatively evaluate the performance of derived global suitability (Hosmer et al., 2013). It is worth to note that here we used the traditional ROC approach to evaluate the suitability surface, which is only one component of the adopted urban CA model in this study, despite that our modeling target is ISA instead of binary urban extent." (Page 7, Line 20-23; Page 8, Line 1-4)*

**Comment #1-5**

Page 7, Section 3.2 (Calibration): What is the time period of the GAIA data used for the model calibration and validation? It's not clear if there was an independent calibration and validation period, or if the calibration/validation were both based on the entire 1985-2015 dataset.

**Response:** thank you for your question. We divided GAIA data into two temporal segments. The time series data of GAIA obtained in periods of 1985-2005 and 2005-2015 were used for calibration and

validation, respectively. We rephrased the title of this subsection and clarified it in our revised manuscript.

*"We calibrated the Logistic-Trend-ISA-CA model at the state level using historical urban extent time series data (i.e., GAIA) from satellite observations (1985-2005)." (page 7, line 19-20)*

*"Furthermore, we validated the model based on the root mean square error (RMSE) (Eq. 5) and coefficient of determination ($R^2$) between the modeled and observed ISAs (2005-2015) at the global scale. That is, we modeled the urban sprawl from 2005 to 2015 using the calibrated Logistic-Trend-ISA-CA model. In general, the relatively low RMSE and high $R^2$ suggest the calibrated urban CA model can capture urban sprawl well." (Page 8, Line 7-10)*

**Comment #1-6**

Page 8, line 1. "Given that the GAIA data were derived from satellite observations with good quality and fine resolution, we harmonized future urban growth trends (2015-2100) from LUH2 under different SSP-RCP scenarios with the derived urban areas from GAIA in 2015." Do the GAIA data and the LUH2 data use the same definition of "urban" land? It may be another reason for the difference between the urban area extents of the two datasets in 2015.

**Response:** thank you for your comments. First, the urban extents from both GAIA and LUH2 database are derived from remotely sensed observations. Specifically, urban extents in LUH2 were initially estimated from spatially explicit built-up area map in 2000 from 1km DISCover dataset (*Loveland et al., 2000*). Although the definition of "urban" in LUH2 and GAIA are similar (i.e., pixel dominated by built-up areas), the GAIA data have a finer spatial resolution (i.e., 30m) and a longer temporal span (1985-2018) at an annual step, with mean overall accuracies of above 90% across different years.

*"Due to the difference of adopted baseline urban extent in each product, there is a distinct gap regarding urban area in these two datasets (i.e., GAIA and LUH2). Specifically, urban extents in LUH2 were initially estimated from spatially explicit built-up data of the Data and Information System Global Land Cover (DISCover) dataset at 1km resolution, which was mainly derived from the Advanced Very High Resolution Radiometer (AVHRR) satellite observations (Loveland et al., 2000; Goldewijk, 2017). While the definitions of "urban" are similar in both products, the differences in urban areas across various regions can be attributed mainly to their spatial resolutions and mapping years. In general, the urban extent in GAIA derived from Landsat has a longer temporal span and a high accuracy, with mean overall accuracies of above 90% across different years (Gong et al., 2020a)." (Page 9, Line 14-21)*

**Comment #1-7**

Page 13, Data availability. This data on fractional urban changes from 2015-2100 will be of much interest to researchers around the world, so I appreciate that you have made the data openly available. Considering all of the data you have generated in this study, another suggestion is that you may want to also share the development probability (Pdev) dataset, which contains the probability of urban

development in each 1km grid cell(?). Using this data, readers could potentially generate their own future urban (fractional) change maps, e.g., based on national urban development/land demand scenarios.

**Response:** thank you for your suggestions. As suggested, we have uploaded the development probability data in FigShare with detailed explanation in our revised manuscript.

*"The gridded dataset of global urban fractional change (2015-2100, 5-year interval) at 1km spatial resolution under eight future development pathways. The global urban development probability map can be downloaded from https://doi.org/10.6084/m9.figshare.20391117.v3 (He et al., 2022)." (Page 15, Line 17-19)*

---

## Author Comment (AC2)

**Reviewer2**

**Comment #2-0**

The paper takes on the very substantial challenge of developing a global dataset of urban fractional changes at a 1km resolution from 2020 to 2100 under eight scenarios of socioeconomic pathways and climate change. The newly developed fractional urban land dataset is quite valuable and is helpful to assess the environmental impact of future urbanization. The manuscript is generally well structured, but the discussion and conclusion need to be reorganized. For example, the conclusion section is too wordy and repeats some information from the method and result sections. Though the method used in this work has been published, I still have some concerns when the model is applied to global-scale modeling. There are some missing details in the method section, and I have provided detailed comments below. Overall, I believe this paper could be publishable after major revision.

**Response:** thank you for your suggestions and comments. We carefully revised the discussion and conclusion part, and added details of the methodology part in our revised manuscript. The detailed point-by-point response is shown below.

**Comment #2-1**

P6, Line 1-2. "We characterized urban fractional change across different states in each country, using the long-term (1985-2015) urban extent data (i.e., GAIA) and the sigmoid growth model." The sigmoid growth model is fitted in the state level, and the estimate parameters (a, b, c, d) are also shown in Fig. 6. How the sigmoid growth model performs in each state, because the urbanization stage is different among developing countries and developed countries. I suggest adding a fit performance of the sigmoid growth model to prove the model reliability and explain the model uncertainty.

**Response:** thank you for your suggestions. We have included the fit performance of the sigmoid growth model across states globally (Fig. S7). First, we fitted the ISA-based urban growth model with calibrated model parameters using the urban extent time series from (1985-2015). Then, we evaluated the performance of sigmoid model using the coefficient of determination ($R^2$) between the modeled and referred ISAs. Our results indicate the sigmoid model can well characterize the urban evolution with $R^2$ above 0.8.

*"Additionally, we evaluated its performance of the ISA-based sigmoid growth model using the coefficient of determination ($R^2$) between the estimated and observed ISAs (1985-2015) at the state level." (Page 8, Line 4-6)*

*"The fit performance (i.e., with $R^2$ above 0.8) of the sigmoid model at the state-level indicates the calibrated model can well characterize the spatially explicit urban growth patterns over past decades using satellite observations (Fig. 7). It is worth to note that there are some states with relatively low performance, probably due to limited increments with different urbanization stages, e.g., in highly urbanized regions or developing regions with low urbanization levels." (Page 12, Line 5-9)*

[Figure]

*Fig 7. The fit performance (R²) of state-level sigmoid model at the global scale. R² is the coefficient of determination between the simulated and referred ISAs over past decades.*

**Comment #2-2**

P6, Line 12-15. "We incorporated the ISA-based growth mechanism with the Logistic-Trend-CA model (He et al., 2023), which incorporates temporal contexts of urban sprawl into the neighborhood configuration…." The sigmoid growth model is estimated at the state level, so did you also train the logistic regression model at the state level? The logistic regression is a binary regression model, but input data (GAIA) is fractional type. How did you implement the model training? I also see the methods in the supplementary materials, so only three spatial proxies (i.e., DEM, Distance to city centers, Distance to major road) are used to training the logistic regression model? And how many samples were selected to train the model in each state?

**Response:** thank you for your comments. We trained the country-specific regression model and obtained the spatially explicit suitability surface map. This is different with the sigmoid growth model because it is too homogeneous for these spatial proxies at the state level. Specifically, we identified those changed and persistent pixels and collected these two sample groups using the stratified sampling strategy. Thereafter, the suitability surface was derived using the regression model based on those spatial proxies, including DEM, slope, distance to city centers, distance to different roads, as well as those land covers. Details of these proxies can be referred in Supplementary Materials. We clarified these issues in our revised manuscript.

*"Specifically, we identified those changed and persistent pixels and collected these two sample groups using the stratified sampling strategy. Thereafter, the suitability surface was derived using the regression model based on those spatial proxies, including the DEM, slope (calculated from DEM), minimum distance to city centers, minimum distance to different grades of roads, as well as various land covers (i.e., evergreen needleleaf forests, evergreen broadleaf forests, deciduous needleleaf forests, deciduous broadleaf forests, mixed forests, open shrublands, savannas, grasslands, permanent wetlands, croplands, cropland/natural vegetation mosaics, urban and built-up lands, permanent snow and ice, barren, water bodies)." (Supplementary Information)*

*"These spatial proxies reflect different spatial aspects related to urban sprawl, such as locations (e.g., minimum distance to major worldwide cities), traffic networks (e.g., minimum distance to major roads, minimum distance to highways, and minimum distance to local roads), terrain (e.g., digital elevation model and slope), were used to train the logistic regression model and land constraints (e.g., protected areas) (Table 1). For example, some spatial proxies (e.g., land cover and protected area) were defined as specific constrains (e.g., suitable or not), while terrain and location proxies were directly calculated from the DEM and distance to the nearest roads (or city centers), respectively." (Page 5, Line 10-16)*

**Comment #2-3**

P7, Line 12. "We calibrated the Logistic-Trend-ISA-CA model at the state level using historical urban extent time series data (i.e., GAIA) from satellite observations". So how did you validate the sigmoid growth model?

**Response:** thank you for your comments. We evaluated the sigmoid growth model by calculating the coefficient of determination ($R^2$) between the simulated and referred ISA time series data. We explained this issue in our revised manuscript. Details can be referred to our response to Comment #2-1.

**Comment #2-4**

P7, Line 13-15. "First, we evaluated the performance of derived suitability surface using the receiver operating characteristic (ROC) method (Sunde et al., 2014). That is, the continuous values can be divided into binary maps using different thresholds to measure the agreement between threshold-derived results and the referenced urban extent." The binary urban land map was used to evaluate the performance of the derived suitability surface. How the thresholds were determined to extract the binary urban maps for each state. Are there large differences in the thresholds among states? If you evaluate the model performance for each state, I suggest that Fig.7 (i.e., model performance at the country level) could show the AUC values at the state level.

**Response:** thank you for your suggestions. First, the receiver operating characteristic (ROC) approach is a threshold-based evaluation approach *(Sunde et al., 2014)*. That is, the continuous values can be divided into binary maps using different thresholds to measure the agreement between threshold-derived results and the referenced urban extent (i.e., identified by their increased ISA during 1985-2005 with a threshold of 0.5) *(Sunde et al., 2014)*. Second, as our response to Comment #2-2, we didn't implement the regression model at the state level, because these spatial proxies are inadequate the heterogeneity spatially at the local scale. Finally, as suggested, we also evaluated the performance of suitability surface at the state-level (Fig. S8). We clarified it in our revised manuscript.

*"We evaluated the performance of derived global suitability using the receiver operating characteristic (ROC) approach, which essentially is a threshold-based evaluation approach (Sunde et al., 2014). That is, the continuous values can be divided into binary maps using different thresholds to measure the agreement between threshold-derived results and the referenced urban extent (i.e., identified by their increased ISA during 1985-2005 with a threshold of 0.5) (Sunde et al., 2014)." (page 7, line 20-23; Page 8, Line 1)*

*"It is worthy to note that here we used the traditional ROC approach to evaluate the suitability surface, which is only one component of the adopted urban CA model in this study, despite that our modeling target is ISA instead of binary urban extent." (page 8, line 2-4)*

*"Furthermore, the suitability surface performs well in most states with relatively high AUC values (Fig S8)." (Page 12, Line 19)*

[Figure]

*Fig S8. The model performance of derived suitability surfaces at the state-level using the indicator of the area under the curve (AUC) at the global scale.*

**Comment #2-5**

P8, Line 5-6. Data harmonization should be cautious because the data source and definitions of GAIA and LUH2-urban are different. It is also simple to use equations (6) and (7) to harmonize the urban land from GAIA and LUH2. As I know, the overlap period for GAIA and LUH2 is 1985-2020. Did the harmonization rate change a lot during the overlap period? I suggest adding an uncertainty analysis for the data harmonization.

**Response:** thank you for your suggestions. First, the urban extents from both GAIA and LUH2 database were derived from remotely sensed observations and the definition of "urban" in LUH2 and GAIA are similar (i.e., pixel dominated by built-up areas). Specifically, the urban extents in LUH2 were initially estimated from spatially explicit built-up area map in 2000 from 1km DISCover dataset (*Loveland et al., 2000*). Although the definition of "urban" in LUH2 and GAIA are similar (i.e., pixel dominated by built-up areas), the GAIA data have a finer spatial resolution (i.e., 30m) and a longer temporal span (1985-2018) at an annual step, with mean overall accuracies of above 90% across different years. Second, to keep the modeled ISA map in the future consistent with satellite derived observations, we chose the ISA map in 2015 as the start point of modeling, instead of the overlap period, for harmonization with LUH2 data. Third, as suggested, we also discussed the uncertainty of the harmonization rate ($\gamma$) in our revised manuscript. We used the quartile coefficient of dispersion (QCD) to measure the range of $\gamma$ over past decades. Our results indicate the QCD is relatively small (0-0.2) in most countries, suggesting the $\gamma$ is robust during the overlap period (Fig. S9). We explained these issues with added discussion in our revised manuscript.

*"Due to the difference of adopted baseline urban extent in each product, there is a distinct gap regarding the urban area in these two datasets (i.e., GAIA and LUH2). Specifically, the urban extents in LUH2 were initially estimated from spatially explicit built-up data of Data and Information System Global Land Cover (DISCover) dataset at 1km resolution, which was mainly derived from the Advanced Very High Resolution Radiometer (AVHRR) satellite observations (Loveland et al., 2000; Goldewijk, 2017). While the definitions of "urban" are similar in both products, the differences in urban areas across various regions can be attributed mainly to their spatial resolutions and mapping years. In general, the urban extent in GAIA derived from Landsat has a longer temporal span and a high accuracy, with mean overall accuracies of above 90% across different years (Gong et al., 2020a)." (Page 9, Line 14-21)*

*"We calculated the QCD using the first (Q1) and third (Q3) quartiles for the annual harmonized rate γ in each country during the overlap period, as Eq.8. The QCD can capture the variation of harmonized rates within each country and is also comparable between countries with different deviation of future projection.*

$$QCD = \frac{Q_3 - Q_1}{Q_3 + Q_1} \tag{8}$$

*where $Q_1$ and $Q_3$ are the first (25th percentile) and third (75th percentile) quantiles of the annual harmonized rate in each country, respectively." (Page 9, Line 22-23; Page 10, Line 1-5)*

*"Our results indicate the QCD is relatively small (0-0.2) in most countries, suggesting the γ is robust during the overlap period (Fig S9). The urban areas in most regions like America, Canada, China, India, and Australia have similar trends in GAIA and LUH2, while their gaps (i.e., QCD) are relatively large in those least developed countries (Fig S9)." (Page 14, Line 5-8)*

[Figure]

*Fig. S9. The quartile coefficient of dispersion (QCD) within each country at the global scale during the overlap period (1990-2015).*

**Comment #2-6**

P8, Line 20-24. Global South countries located in middle Asia, south America, and Africa, would likely experience more noticeable urban growth than Global North countries in the future, e.g., the growth rates of the United States of America (USA) and China are 3.95 and 1.05 times in 2100 under SSP2-RCP4.5, respectively, relative to the base year of 2015. These two sentences are contradictory, you made me confused.

**Response:** thank you for your comments. We improved this sentence in our revised manuscript.

*"Among these scenarios, Global South countries located in middle Asia, south America, and Africa, would likely experience more noticeable urban growth than Global North countries in the future (Fig. 2), i.e., the average growth rate (i.e., 2100/2015) of Global South countries is around 5 under various future scenarios, while the average growth rate in Global North countries is mostly lower than 3." (Page 10, Line 9-12)*

**Comment #2-7**

P12, Section 4.3. In this section, you mainly compared the spatial pattern of the newly developed urban fractional dataset and previous datasets. I suggest adding a comparison analysis of future urban land area between the harmonized data and other available datasets.

**Response:** thank you for your suggestions. Although the overall trends of future global urban sprawl in our results are similar as those two global urban extent datasets (*Chen et al., 2020; Gao et al., 2020*) (*Fig. S10*), their magnitudes are notably different. This is mainly attributed to the variation caused by the urban area growth estimation model adopted in different products. For instance, the temporal trend of our results was mainly inherited from LUH2, which essentially was estimated from multiple integrated assessment models. However, for products in Chen et al. (2020) and Gao et al. (2020), their urban areas were estimated using panel analysis and data-driven approaches, respectively, based on four-epoch time series data of global human settlement layer. We clarified it in our revised manuscript.

*"The overall trends of future global urban sprawl in our results are similar as those two global urban extent datasets (Chen et al. (2020a); Gao and O'neill (2020)), but their magnitudes are notably different (Fig. S10). These deviations are mainly attributed to the variation caused by the urban area growth estimation model adopted in different products. For instance, the temporal trend of our results was mainly inherited from LUH2, which essentially was estimated from multiple integrated assessment models. However, for products in Chen et al. (2020) and Gao et al. (2020), their urban areas were estimated using panel analysis and data-driven approaches, respectively, based on four-epoch time series data of Global Human Settlement Layer (GHSL)." (Page 14, Line 15-23; Page 15, Line 1-4)*

[Figure]

*Fig. S10. The projected urban area of our results with two similar global urban datasets from Gao and O'neill (2020) and Chen et al. (2020a) under all future scenarios across years.*

**Comment #2-8**

Fig. S6. I note that there will be no low-density ISA area in the city you show after 2060, and it seems that most of the urban area have the same ISA fraction. It also existed in other metropolitan areas (e.g., Fig. 10 and 11, New York city). so, I suggest to explain why.

**Response:** thank you for your comment. The projected urban area in LUH2 data will decrease in most regions due to the population decrease, especially after 2050. With the widely adopted assumption that it is irreversible for the transition from urban to non-urban, here we assumed the urban area growth would be plateaued once it reaches the peak, so do the ISA. Illustrated as the Georgia state in the US (e.g., Atlanta in Fig. S6), the total urban area reaches the peak after 2080 (Fig R1). We explained it in our revised manuscript.

*"Given that the conversion from non-urban to urban is commonly assumed to be irreversible (Li et al., 2015), we assumed that the urban area in regions with projected population decline in the future would plateau after reaching its peak." (page 9, line 7-9)*

[Figure]

*Fig. R1. The temporal trend of urban area growth in the state of Georgia.*

---

## Author Response (AR2)

**Reviewer 1**

**# Comment 0**

My concerns were well addressed by the authors. Thank you to the authors for your revisions, and for providing the additional information and data requested. The new dataset included with the revised submission ("global spatially explicit map of urban development probability") will be another useful resource. I recommend to accept the paper.

**Response:** thank you for your positive comments and suggestions to improve our manuscript.

**Reviewer 2**

**# Comment 0**

The manuscript describes the development details of a future fractional urban impervious surface area (ISA) dataset for 2015-2100 at a 5-year interval. I think the newly developed future urban ISA dataset will be very useful for understanding the impact of future urbanization on the ecosystem. I have reviewed the revised manuscript and the point-by-point responses to the comments. The authors have revised the manuscript following the suggestions and comments closely. They did a lot of work in quantifying the uncertainties of data harmonization, which increased the reliability of the model and dataset. Overall, the authors have done a good job in addressing these comments, and the manuscript has been improved a lot. But I still have several small suggestions and provide them in the specific comments.

**Response:** thank you. As suggested, we have carefully revised our manuscript and provided a detailed point-by-point response below.

**# Comment 1**

P4, Line 16-19. "Given that there are currently no urban fractional ISA dataset in high spatial resolution (e.g., 1km) directly obtained from satellite observations, here we adopted the commonly used strategy through spatial aggregation from high-resolution (e.g., 30m) urban extent data to derive the ISA time series data for modeling."

As I know, there are at least two fractional impervious surface area datasets have been developed. For example, the Global Man-made Impervious Surface (GMIS) developed by NASA Socioeconomic Data and Applications Center could be available since 2017, which can be accessed at https://sedac.ciesin.columbia.edu/data/set/ulandsat-gmis-v1.

**Response:** thank you for this great suggestion. We agree that the GMIS data are fractional products with detailed information on impervious surfaces within each 30m grid. However, as we stated in our manuscript, the temporal dynamics of urban fractional information derived from satellite observations are crucial to our model development with improvements, which needs to be improved in the GMIS data due to its one epoch (i.e., 2010). As such, we clarified this issue and rephrased our descriptions in our manuscript as below.

*"Probably due to the absence of long-term and fine resolution annual global urban extent time series data (Li et al., 2015; Shi et al., 2017; Song et al., 2016; Brown De Colstoun et al., 2017), characterizing the temporal pattern of urban sprawl dynamics has not been comprehensively explored, in particularly coupling with urban CA models. Although urban fractional data with detailed impervious surfaces have been developed recently, such as the Global Man-made Impervious Surface (GMIS) data (Brown De Colstoun et al., 2017),*

*information of long-term urban fractional dynamics is still highly required for urban CA model improvement (Page 3, Line 17-22).*

*"Given that there are currently no long-term urban fractional (i.e., ISA) dynamic products in high spatial resolution (e.g., 1km) directly obtained from satellite observations (Brown De Colstoun et al., 2017), here we adopted the commonly used strategy through spatial aggregation from high-resolution (e.g., 30m) urban extent data to derive the ISA time series data for modeling." (Page 4, Line 20-23)*

**# Comment 2**

P7, Line 15, the stochastic disturbance item is missed in equation (4), and no description of the 'SP' item.

**Response:** we are sorry about it. As suggested, we have included the stochastic disturbance item (SP) in Eq. (4) in our revised manuscript, with details provided in our Supplementary Texts.

$$"P_{dev} = P_{suit} \times \Omega \times Land \times SP \qquad (4)$$

*where $P_{dev}$ indicates the development probability; $P_{suit}$, $\Omega$, $Land$, and $SP$ represent the suitability surface, neighborhood, land constraint, and stochastic disturbance, respectively. Details of these parameters can be referred to in the Supplementary texts." (Page 7, Line 19-22)*

**# Comment 3**

P9, Line 4-6. "Here we assumed the trend of urban sprawl at the state level is consistent with that at the country level, as population and GDP change are commonly estimated at the country and regional scale".

This is a simple downscaling method to get the future urban land area demand of each state, and may result in some uncertainties as the urbanization stage varies. It is also contradictory to the description in the first paragraph of section 3.1, indicating the urbanization stages information was not used in the future urban land area prediction. The better way to downscale the future urban land area from country to state is to set the urbanization stage as a weight.

**Response:** thank you for these valuable suggestions. The modeling of future urban dynamics includes two components: 1) urban area estimation within a given administrative unit and 2) spatial fractional growth of urban extent. As stated in the first paragraph of Section 3.1, we characterized different urban growth patterns at the state level, considering their varying urbanization stages. This is helpful to deepen our modeling mechanism at the grid scale with gradual ISA change. Whereas for the future urban land area, we directly harmonized the trend gained from the integrated assessment model (IAM) under diverse SSP-RCP scenarios, where the urban area was commonly estimated by the population and GDP without explicitly considerations of urbanization stages (*Hurtt et al., 2011*). As suggested, this factor could be considered in future urban land area estimation by weighting the urbanization stages. We discussed this issue in our revised manuscript.

*"It is worthy to note that here we directly inherited the future trend of urban areas from the integrated assessment model (IAM) under diverse SSP-RCP scenarios (Hurtt et al., 2011) across different states in each country, harmonized with historical urban extent dynamics*

*from satellite observations. However, the urbanization stage was not considered in those IAM models, which were mainly driven by demographical and socioeconomic factors. In the future, the urbanization stages could be a weight factor when downscaling urban areas from country to state." (Page 14, Line 18-22)*

**Comment 4**

One of the corresponding authors published related work in 2019 and 2021 (Li et al., 2019; Li et al., 2021). The two papers also simulated the future urban land expansion based on the nightlight data derived urban land. So, what are the improvements of the newly developed dataset compared with previous work? It can be included in the discussion.

**Response:** thank you. Compared to our previous work (*Li et al., 2019; Li et al., 2021*), this study developed an ISA-based urban CA model that considers long-term temporal contexts of urban evolution from satellite observations, enabling urban fractional modeling at the grid scale. Such a modeling mechanism is quite different to those studies with binary conversions. As suggested, we highlighted it in our revised manuscript.

*"Compared to other global urban products under future scenarios, our results can promote future urban land use efficiency by simulating gradual urban fractional change with notably improved spatial details (i.e., 1km) (Li et al., 2021; Li et al., 2019a; Gao and O'neill, 2020; Chen et al., 2020a)." (Page 17, Line 9-11)*

**Comment 5**

Same as Reviewer 2, Fig. S6. I note that there will be no low-density ISA area in the city you show after 2060, and it seems that most of the urban area have the same ISA fraction. It also existed in other metropolitan areas (e.g., Fig. 10 and 11, New York city).

This may be resulted from the spatial allocation algorithm. Specifically, the grid cell with high suitability always has more ISA increment. On the other hand, no enough newly developed urban land grid cells to allocate the increased ISA. Thus, there should be a balance between urban land expansion and ISA increase in the existing urban land pixels. It will be good to improve the spatial allocation model by constraining the filling of urban inner space and expansion of urban bound. Thus, there should be some discussions about the uncertainties of the spatial allocation model.

**Response:** thank you for raising this comment. Yes, we acknowledged this effect, whereas it can be explained from the following two aspects. On the one hand, urban growth in these cities (e.g., Atlanta) almost plateaued after 2060, primarily determined by the trend in LUH2. On the other hand, although most pixels in the urban fringe areas show similar ISA values (i.e., almost the same color), their values are different regarding the ISA gain within each period (Fig. R1). In general, the ISA-based growth in these cities (e.g., Atlanta) was also related to the calibrated state-specific Sigmoid growth curve, which was determined primarily by the long-term urban extent time series from satellite observations.

[Figure]

*Fig. R1. The distribution of ISA increase within a given period (a) and the corresponding ISA value at the end year (b) in Atlanta (US), under the middle of the road (SSP2-RCP4.5).*

As suggested, we also discussed the uncertainties of the spatial allocation model in our revised manuscript. It is worth noting that grids with higher ISA increments were mainly determined by the suitability values and the urbanization level (i.e., indicated by ISA) during the iteration. For instance, grids with relatively high ISA values are associated with lower growth rates than those at the Sigmoid curve's middle stage. Also, although we introduced the stochastic term to promote new seeds of urban development in our model, the probabilities for those non-urbanized pixels are relatively low for development in the future, given that the urban growth is relatively slow when the ISA value is very low (i.e., close to zero). We clarified this issue in our revised manuscript as below.

*"In addition to the suitability, the state-based trend of ISA growth from satellite time series data may also impact the ISA growth at the pixels, particularly for those with extremely low and high ISA values. It's worth noting that the ISA-based growth in our modeling mechanism may underestimate the growth of pixels with very low ISA values or non-developed, although the stochastic disturbance term has been involved in our modeling mechanism. Meanwhile, the rate of urban fractional growth is slow for pixels around the city centers with relatively high ISA values. Appropriate strategies by constraining the filling of urban inner spaces and the expansion of urban bound should be developed to improve the spatial allocation of urban CA model." (Page 11, Line 15-21).*

**#Reviewer 3**

**# Comment 0**

The authors have developed a global urban fraction change dataset with a resolution of 1 km from 2020 to 2100 (with 5-year intervals), covering eight socio-economic development pathways and climate change scenarios. The researchers used an S-shaped growth model and

annual Global Artificial Impervious Area (GAIA) data to describe the ISA growth pattern over the past few decades (i.e., 1985-2015). By combining the ISA-based growth mechanism with the CA model, the researchers quantitatively evaluated state-specific urban CA models on a global scale. This method can capture the spatially explicit changes in ISA and the gradual ISA changes within pixels, which is very useful for supporting quantitative analysis of ecological and environmental changes caused by urbanization at a fine scale.

Overall, this paper provides a novel and practical approach and a valuable dataset for studying global urbanization. The authors have provided detailed descriptions of their data and methods, and have provided ample evidence to support their conclusions. Therefore, I believe this paper can be accepted.

**Response:** thank you for the positive comments. As suggested, we have carefully checked the missing statement in our updated manuscript.

**# Comment 1**

However, there is a minor spelling issue that needs to be corrected. I could not find the definition of SP in Eq. 4. Please check this issue.

**Response:** thank you for your suggestions. We have included the definition of the stochastic disturbance item (SP) in equation (4) in our revised manuscript and more details of these spatial parameters can be found in the Supplementary texts.

$$"P_{dev} = P_{suit} \times \Omega \times Land \times SP \qquad (4)$$

*where $P_{dev}$ indicates the development probability; $P_{suit}$, $\Omega$, Land, and SP represent the suitability surface, neighborhood, land constraint, and stochastic disturbance, respectively. Details of these parameters can be referred to in the Supplementary texts." (Page 7, Line 19-22)*